# Fibrillarin Ribonuclease Activity is Dependent on the GAR Domain and Modulated by Phospholipids

**DOI:** 10.3390/cells9051143

**Published:** 2020-05-06

**Authors:** Francisco Guillen-Chable, Ulises Rodríguez Corona, Alejandro Pereira-Santana, Andrea Bayona, Luis Carlos Rodríguez-Zapata, Cecilia Aquino, Lenka Šebestová, Nicolas Vitale, Pavel Hozak, Enrique Castano

**Affiliations:** 1Biochemistry and Molecular Plant Biology Department, Centro de Investigación Científica de Yucatán, A.C. Calle 43 No. 130, Colonia Chuburná de Hidalgo, Mérida C.P. 97200, Yucatán, Mexico; francisco.guillen@cicy.mx (F.G.-C.); Ulises.RodriguezCorona@ircm.qc.ca (U.R.C.); andrea.bayona.her@gmail.com (A.B.); cecilia.aquino-perez@img.cas.cz (C.A.); 2Industrial Biotechnology Department, Centro de Investigación y Asistencia en Tecnología y Diseño del Estado de Jalisco, A.C., Camino Arenero 1227, el Bajio, Zapopan C.P. 45019, Jalisco, Mexico; apereira@ciatej.mx; 3Dirección de Cátedras, Consejo Nacional de Ciencia y Tecnología, Av. Insurgentes Sur 1582, Alcaldia Benito Juarez C.P. 03940, Ciudad de Mexico, Mexico; 4Biotechnology Department, Centro de Investigación Científica de Yucatán, A.C. Calle 43 No. 130, Colonia Chuburná de Hidalgo, Mérida C.P. 97200, Yucatan, Mexico; lcrz@cicy.mx; 5Department of Biology of the Cell Nucleus, Institute of Molecular Genetics of the CAS, v.v.i., Videnska 1083, 142 20 Prague, Czech Republic; lenka.sebestova@img.cas.cz (L.Š.); pavel.hozak@img.cas.cz (P.H.); 6Faculty of Science, Charles University, Albertov 6, 128 00 Prague, Czech Republic; 7Institute of Celullar and Integrative Neuroscience (INCI), UPR-3212 The French National Centre for Scientific Research & University of Strasbourg, 67000 Strasbourg, France; vitalen@unistra.fr

**Keywords:** nucleolus, ribonucleolar particle, rRNA, fibrillarin, phosphoinositides, viral progression

## Abstract

Fibrillarin is a highly conserved nucleolar methyltransferase responsible for ribosomal RNA methylation across evolution from Archaea to humans. It has been reported that fibrillarin is involved in the methylation of histone H2A in nucleoli and other processes, including viral progression, cellular stress, nuclear shape, and cell cycle progression. We show that fibrillarin has an additional activity as a ribonuclease. The activity is affected by phosphoinositides and phosphatidic acid and insensitive to ribonuclease inhibitors. Furthermore, the presence of phosphatidic acid releases the fibrillarin-U3 snoRNA complex. We show that the ribonuclease activity localizes to the GAR (glycine/arginine-rich) domain conserved in a small group of RNA interacting proteins. The introduction of the GAR domain occurred in evolution in the transition from archaea to eukaryotic cells. The interaction of this domain with phospholipids may allow a phase separation of this protein in nucleoli.

## 1. Introduction

The nucleolus is a well-studied nuclear structure which has been shown for a long time to separate into three components depending on the particular step of ribosome biogenesis: the fibrillar center (FC) where RNA pol I is found, the dense fibrillar component (DFC) where protein fibrillarin is concentrated, and the granular component (GC) that is enriched in nucleophosmin (NPM1) [1,2]. The nucleolar proteome consists of hundreds of different proteins, as well as a large set of different types of RNA that need to be in a particular compartment in order to carry their function [3]. All these components form a multi-layer core found in a liquid gel structure. The nucleolar assembly is controlled in a concentration-dependent phase transition, as shown earlier by [4], where the nucleoli behave as multi-component liquid-phase droplets ruled by thermodynamic forces [5]. An elegant set of experiments using a microfluidic device showed the phase dynamics of fibrillarin and other nucleolar components [6]. Furthermore, the composition, morphology, and fusion dynamics of nucleoli are altered by ATP depletion, as shown using fibrillarin-GFP in a lapse movie. Thus, fibrillarin condensates into dozens of small droplets throughout the nucleus [7]. Transcription of pre-ribosomal RNA at active nucleolar organizer regions (NORs) lowers the critical concentration for phase separation of nucleolar proteins, such as fibrillarin and NPM1, and nucleates the assembly of the fibrillarin-rich DFC [8]. The spatial separation of the nucleolar sub-compartments is dictated by differences in their viscoelastic properties, especially in the surface tensions of the fibrillarin- and NPM1-rich phases with respect to the surrounding nucleoplasm [9]. 

Fibrillarin in eukaryotic organisms contains a glycine/arginine-rich domain (GAR), a spacer region called BCO, a central domain with RNA binding motif, and an alpha helix domain [10,11]. Fibrillarin is an essential 2’-O-methyltransferase for all eukaryotes, mediating the methylation of rRNA and histone H2A at a glutamine in position 104 in humans and 105 in yeast required for the epigenetic regulation of ribosomal genes [12,13]. Fibrillarin is important in ribosome biogenesis and also plays a role in other processes, including viral progression. RNA viruses like coronavirus, infectious bronchitis virus (IBV), influenza, and human immunodeficiency virus (HIV) require fibrillarin for cell to cell infection, both in plants and humans [10,14,15,16]. Fibrillarin is known to associate with different nuclear and nucleolar RNAs (U3, U8, U13, U14, U60, x, y, snR3, snR4, snR8, snR9, snR10, snR11, snR30, snR189, and snR190) [1,17] and other proteins to form a variety of snoRNPs. The best-characterized and most studied snoRNP is composed of the small nucleolar RNA (snoRNA) U3. snoRNA U3 is involved as a guide RNA in the initial steps of pre-rRNA processing [18] and cleavage that leads to the maturation of 18S rRNA [19]. For this early processing, several factors, such as fibrillarin, nucleolin, and the U14 snoRNA, are required [20,21,22]. It has been suggested that liquid–liquid phase separations (LLPSs) are implicated in synthesis and processing modifications during ribosomal production. Phase separation has been observed in proteins that contain intrinsically disordered regions (IDRs) in their sequences [23,24], i.e., fibrillarin, nucleolin, and Gar1 proteins, contain IDR. LLPS in this process ensures a membrane-less environment for the production and highly ordered process of ribosome assembly. Earlier results indicate that fibrillarin can bind to phosphatidylinositol 4,5-bisphosphate (PIP2). Lipids play crucial roles in nuclear function and dynamic architecture, and fibrillarin is involved in the lipid phase and RNA binding in the cell [25,26,27]. 

Phosphoinositide pathways are regulated by several enzymes in both cytosol and the nucleus. Each of the seven different phosphorylated phosphoinositides [28] is known to localize in different parts of the cell [29]. Therefore, a particular environment for these phosphoinositides, together with their partners, can be in a particular phase that can be differentiated by metabolic changes in the lipids. This hypothesis involves a specific localization for a protein–lipid complex together with a particular function that can be altered by lipid metabolism and thus results in a re-localization of the components or alteration of complex formation, leading to the movement of some components due to the biochemical properties of the new lipids. We are still far from showing that RNA, lipids, and proteins can organize in particular structures that can phase separate by means of lipid metabolism and posttranslational protein modifications. However, such processes would reduce the energy requirements needed for cells to organize particular structures and to relocate particular complexes upon phosphorylation changes in the lipids. Dynamic proteins like fibrillarin are intriguing subjects to test as they are known to exist in particular phase separation. However, their link to structural changes by RNA alteration suggests that this component is also essential for proper localization, as during RNA inhibition [30]. Therefore, fibrillarin localization is not a simple phase separation that requires a protein–lipid complex, but also RNA to have a functional role through ribosome biogenesis. 

Here we show that fibrillarin has a ribonuclease activity for rRNA, and this general activity is blocked when it is bound to snoRNA U3. The ribonuclease activity of fibrillarin is carried by the GAR domain, which is also responsible for interaction with different phosphoinositides. Moreover, we also show that the ribonuclease activity of fibrillarin increases with phosphatidylinositol 5-phosphate (PI5P) and reduces in the presence of phosphatidic acid (PA) or during binding to Guide RNA U3. Ribonucleases are involved in vital cellular functions, including cytoplasmic and nuclear RNA degradation, RNAi, antiviral defense, DNA synthesis, and RNA processing [31,32,33,34,35]. Pre-rRNA maturation involves the activity of specific RNases and RNA modifying enzymes to achieve functional rRNA [36,37]. Most pre-rRNA processing is mediated by the co-transcriptional association of nascent transcripts with ribosomal proteins (RP) and several small nucleolar ribonucleoprotein particles (snoRNP) containing fibrillarin [38]. 

## 2. Materials and Methods

### 2.1. Cell Lines, Cell Culture, and Transfection Assays

HeLa cells were cultured as previously published [25]; transient transfection was performed at 80% confluence using polyethylenimine (PEI) with 10 µg of plasmid DNA in 1 mL of Dulbecco´s modified Eagle medium (DMEM) without fetal calf serum. Distribution patterns of the wild type and fibrillarin mutants, coupled with GFP reporter, were analyzed in 200 cells for at least three separate experiments. Plasmid DNA for transfections was purified from bacterial cultures using maxiprep columns (QIAGEN, Hilden, Germany).

A stable SNAP-tag-Fib cell line was generated as follows: transfection with PEI was performed in an 80% confluence 60-mm plate of U2OS osteosarcoma cell line. The cocktail for transient transfection (1.5 µg of pfSNAP-tag-fibrillarin plasmid DNA and 9 µL of PEI) was incubated for 15 min at RT and added dropwise. After 48 hours of transfection, 1000 µg/mL of geneticin G-418 of Sigma-Aldrich® was added and incubated for 9 days. The selection of positive transfected colonies was made after washing the dead cells with PBS and changing the culture media. Cells were stained with a fluorescent substrate that can be used to label SNAP-tag® fusion proteins, and positive colonies were selected prior to use in a bigger culture. The fluorescent substrate used for this work was SNAP-Cell® Oregon Green®. Specific primary anti-PIP2 from Echelon™ (Z-A045) was used for immunofluorescence detection of PIP2 and labeled by anti-mouse IgM secondary antibody conjugated with Alexa Flour® 555 from Life Sciences. The PA sensor Spo20p-GFP was described previously [39].

### 2.2. Microscopy 

Wide-field microscopy was performed on a Leica DM6000 (Leica Microsystems, Wetzlar, Germany; filter cubes: DAPI (Ex: 360/40, Em: 470/40), FITC (Ex: 480/40; Em: 527/30), TRITC (Ex: 546/12; Em: 600/40), Cy5 (Ex: 620/60; Em: 700/75)) using an HCX PL APO 100x/1.40-0.70 OIL objective, a Leica EL6000 with an HXP 120W/45C Vis Hg light source, Type F immersion liquid (Leica Microsystems), a Leica DFC350 FX camera, and Las X software.

### 2.3. Structured Illumination Microscopy 

Images were acquired using the 3D-SIM system DeltaVision OMX (GE Healthcare Life Sciences, Marlborough, MA, USA) with PLAN APO N 60x/1.42 OIL objective, a 4x pco.edge 5.5 sCMOS camera and DeltaVision OMX (Buckinghamshire, UK) controlling software. Lasers with wavelengths 405, 488, 568, and 642 nm with filters for DAPI (Ex: 395.5/29; Em: 435.5/31), FITC (Ex: 477/32; Em: 528/48), and Alexa Fluor 568 (Ex: 571/19; Em: 609/37). Immersion oil laser liquid (Cargille Laboratories, Cedar Grove, NJ, USA; Code: 5610; n = 1.5160). Image reconstruction and registration were processed with DeltaVision softWoRx 6.5.2. software. Wiener filters were set to 0.001. Samples were mounted to 90% glycerol supplemented with 5% N-propyl gallate.

### 2.4. Cloning

Total RNA from U2OS cells was extracted with RiboZol^TM^ (Solon, OH, USA), following the manufacturer’s instructions. The sequences from fibrillarin, GAR1, and SLM14 were obtained using cDNA from the reverse transcriptase strategy of TaqMan^TM^ and amplified by DreamTaq^TM^ polymerase (Thermo Fisher Scientific, Waltham, MA, USA), following the provider’s instructions. The gene encoding fibrillarin (NM_001436.3) was amplified by RT–PCR from total RNA extraction of HeLa cells and cloned into the NcoI and BamHI sites of the pET-15b vector (Fib_Fwd 5′-CCATGGATGAAGCCAGGATTCAGTCCCCGTG-3′; Fib_Rev 5-GGATCCTCAGTTCTTCACCTTGGGGGGTGGC-3′). For protein expression in HeLa cells, the fibrillarin sequence was cloned into the XhoI and BamHI sites of the pEGFP-N1 vector to generate pGFP-FIB. The N-terminal part (amino acids 1 to 134) containing the GAR domain of the fibrillarin was amplified from the pET15b: HsFib vector (GB_forward 5′-CCATGGATGAAGCCAGGATTCAGTCCC-3′; GB_reverse 5′-CTCGAGGTACTCAATTTTGTCATCTCCTTCC-3′, and cloned into the NcoI and XhoI sites of the pET42b vector). Once cloned into pET42b, small fragments from fibrillarin were obtained by complete plasmid PCR mutagenesis with the following primers: BCO (Forward 5′-AGAATGTGATGGTGGAGCCGCA-3′, reverse pET42b 5′-CCATGGACCCGCGTCCCTCAA-3′); HsGAR domain (Forward pET42b 5′-CTCGAGCACCACCACCACCA-3′; reverse 5′-TCTTCCTCCTCCTCCACCGCC-3′; and miniGAR (Forward pET42b and reverse 5′-TCACCAAAGCCCCCTCGGCC-3′).

The specific primers used for cloning the GAR1 and GAR domains from Lsm14 genes were FwGAR1 5′-CCATGGATATGTCTTTTCGAGGCGGAGG-3′, and RvGAR1 5′-CTCGAGATGTCCTCTCCCTCTGAAACC-3′ between NcoI and XhoI restriction sites, respectively. For GAR-Lsm14, the domains were Fw 5′-AAGAATTCGATGACAATAGAGAA-3′ and Rev 5′-TAAGCTTAG GGTCCAAAAGCTGTGCTGT-3′ between EcoRV and HindIII restriction sites, respectively.

### 2.5. Recombinant Protein Expression and Purification

Recombinant fibrillarin was expressed in *Escherichia coli* BL21 gold with 1 mM isopropyl-D-1-thiogalactopyranoside (IPTG) at 25 °C for 5 h. Harvested cells were resuspended in protein extraction buffer (500 mM NaCl, 25 mM tris pH 8, 10% glycerol, 20 mM imidazole, 0.1% Tween 20, 0.1 mM AEBSF, and 0.1 mM DTT) and broken down by sonication. After clarification by centrifugation (17,400× *g* for 15 min), the supernatant was loaded onto a Ni-NTA agarose column (Thermo Fisher) and washed three times with the extraction buffer and then eluted with a linear gradient from 70 to 200 mM imidazole in BC-100 buffer (20 mM Tris-HCl buffer, pH 8, 100 mM NaCl, 0.2 mM EDTA, 10% glycerol) and revised by 15% SDS-PAGE. The fraction containing fibrillarin was passed through MonoQ sepharose (Amersham Pharmacia, Buckinghamshire UK), utilizing a 0.1 to 0.5 KCl gradient to elute the fibrillarin. Fibrillarin containing fractions were pulled and dialyzed against BC-100 and 0.1 mM AEBSF. After the MonoQ sepharose purification step, the fraction containing fibrillarin was loaded on a MonoS (Amersham Pharmacia) column resin and eluted with a linear gradient from 0.1 to 0.5 M KCl in BC-100 buffer with 0.1 mM AEBSF. The purity of proteins was revised by 15% SDS- PAGE, followed by silver staining. For peptides cloned into pET42b vector, the supernatant was first loaded into a Ni-NTA agarose column followed by loading into a glutathione–sepharose column and eluted with BC-100 buffer with 10 mM of reduced glutathione and 0.1 mM of AEBSF.

*Homo sapiens* GAR1, Lsm14, HsGAR-Fib, and the viral protein TGB1 were expressed in *E. coli* Rosetta competent cells. The recombinant production of these proteins was induced with 1 mM IPTG at 25 °C for 4 h. Induced cells were harvested by centrifugation at 4000 × *g* for 20 min at 4 °C, resuspended in lysis buffer (50 mM Tris-HCl pH 8, 300 mM NaCl, 20 mM Imidazole, 10% glycerol, 0.1% of Triton X-100) and supplied with 0.1 mM AEBSF and 0.1 DTT as a protease inhibitor to reduce respective agents, then sonicated 10 times with 30 s ON/30 s OFF cycles. The fragmented cells were clarified by centrifugation at 15000 × *g* for 20 min at 4 °C, and the supernatant was clarified for the IMAC purification strategy, using 100 µL of nickel beads (Thermo Fisher^TM^) and incubated for 30 min at 4 °C in a rotor. The column was washed using 10 volumes of beads lysis buffer, along with an increased concentration of NaCl from 100 to 500 mM. The proteins were eluted with 50 mM-Tris-HCl pH 8, 100 mM NaCl, and 250 mM imidazole and supplied with 10% glycerol, 0.1 mM AEBSF, and 0.1 mM DTT. All purification steps were done at 4 °C to reduce proteolysis. Proteins were stored at −80 °C until use in the next experimental procedures.

### 2.6. Exponential Megaprimer PCR (EMP) Strategy to Introduce the GAR Domain Coding Region into RNP Complex

Following the EMP strategy [40] to introduce long DNA sequences into plasmids, we cloned the N-domain of fibrillarin into the pLink plasmid containing the coding sequences for NOP56/NOP58, 15.5K, previously reported by [41], owed to the fact that the coding region of fibrillarin is truncated in amino acid 83, i.e., lacking the GAR domain coding sequence.

For the GAR domain primer synthesis, we used the following primers: FW1-GARFib-EMP: 5′- ATGAAGCCAGGATTCAGTCC-3′ and RV1-EMP: 5′- GGACTGAATCCTCGCTTCATCACATTCTTCCCCGACTGGT-3′ in a reaction containing 1X HF Phusion buffer (Thermo Fisher, CAT F530S), 200 µM of each dNTP, 0.5 µM primer FW1, 0.5 µM primer RV1, 25 ng of pET15b:fibrillarin (with full sequence) as plasmid DNA template, and 0.02 U/µL Phusion DNA polymerase (Thermo Fisher) in a final volume of 50 µL. Following this, PCR conditions were 98 °C, 30 s as an initial denaturing step, followed by 25 cycles of denaturing (98 °C, 10 s), annealing (63 °C, 30 s), and extension (72 °C, 15 s). The 266-bp product of the first PCR was analyzed by agarose gel 1% electrophoresis and purified with the QIAquick Gel Extraction Kit (Qiagen, Hilden, Germany, ID: 28704).

The GAR domain DNA region was introduced with a second PCR using the primers FW1, RV2-GARFib-EMP: 5′-GGATCCCATAGTTAATTTCT-3′, and the PCR product from the first PCR (as a mega primer) in a reaction containing 1X HF Phusion buffer, 200 µM of each dNTP, 0.5 µM primer FW1, 0.5 µM RV2, 25–50 ng mega primer, 25 ng of megaplasmid as a template, and 0.02 U/µL Phusion DNA polymerase (Thermo Fisher^TM^) in a final volume of 50 µL. The PCR conditions were an initial denaturing step at 98 °C for 30 s, 25 cycles, with a denaturing step (98 °C, 10 s), an annealing step (56 °C for 30 s), and an extension (72 °C for 30 s), with no final extension step.

### 2.7. In Vitro Ligation and Transformation of the EMP Product

Using KLD Enzyme Mix from NEB (Cat M0554S), the PCR product from the second reaction was phosphorylated, ligated, and the template DNA removed with kinase, ligase, and DpnI enzymes contained in enzyme mix. The following components were used in a 10-µL reaction: 1X KLD reaction buffer, 1 µL KLD enzyme mix, 1–4 µL EMP product, and nuclease-free H20 up to 10 µL. The reaction was incubated at 25 °C for 5 min, with 5 µL of the resulting product used for transformation into 50 µL chemically competent *E. coli* Top 10 cells.

### 2.8. Recombinant RNP Complex Expression and Purification

Ribonucleoprotein complexes (RNP), with or without the GAR domain sequence, were expressed using *E. coli* BL21 DE3 competent cells. The induced cells were harvested by centrifugation at 4000 × *g* for 20 min at 4 °C, resuspended in lysis buffer (50 mM Tris-HCl pH 8, 300 mM NaCl, 20 mM Imidazole, 10% glycerol, 0.1% of Triton X-100), and supplied with 0.1 mM AEBSF and 0.1 DTT as protease inhibitor and reducing agent, respectively. The fragmented cells from sonication were clarified by centrifugation at 15000x *g* for 20 min at 4 °C, and the supernatant used for the IMAC purification strategy, as stated above. Imidazole elution was carried out with 50 mM-Tris-HCl pH 8, 100 mM NaCl, 10% glycerol, and 0.1 mM AEBSF and 0.1 mM DTT. MonoQ purification was carried out to remove the remaining contaminants. All purification steps were done at 4 °C to reduce proteolysis. The proteins were stored at −80 °C until use in experimental procedures.

### 2.9. SNAP-Tag-Fibrillarin Purification from HeLa Cells

The media from the HeLa cell cultures was discarded, and the cell monolayer washed twice with cold PBS and the excess discarded. Then, 500 µL of buffer A (10 mM HEPES pH 7.9, 10 mM KCl, 0.1 mM EDTA, 0.4% NP-40, 0.5 mM AEBSF, and 0.1 DTT) was added to each plate, the cells were scraped and resuspended with up- and down-pipetting, transferred to a fresh 1.5-mL tube and incubated for 10 min at room temperature, then centrifugated at 15,000 × *g* at 4 °C for 5 min. The cytoplasmic upper layer (supernatant) was stored at −80 °C. The pellet was resuspended in 150 µL of buffer B (20 mM HEPES pH 7.9, 0.4 M NaCl, 1 mM EDTA, 10% glycerol, 0.5 mM AEBSF, and 0.1 mM DTT) and placed on ice for 2 h, vortexing every 15 min, and then centrifugated at 15,000 × *g* at 4 °C for 5 min; the upper layer was stored at −80 °C as the nuclear protein fraction. Proteins were visualized in SDS-PAGE. In-vitro labeling with SNAP-Biotin strategy was done following the NEB manufacturer’ instructions in a reaction tube: 1 mM DTT, 10 µM SNAP-tag substrate, 600 µL nuclear protein extract, and volume adjusted with PBS to 800 µL with 0.5 mM AEBSF. The tube was incubated at 37 °C for 30 min and stored for purification steps. Biotin–streptavidin magnetic beads were used for purification, following the manufacturer’s instructions (Thermo Fisher Scientific).

### 2.10. Western Blot Assay

Fibrillarin was loaded in 12% acrylamide gel to perform SDS-PAGE, then transferred to a nitrocellulose membrane and blocked with 3% of BSA in PBS at room temperature for 1 h. The membrane then was incubated with rabbit polyclonal anti-Fib antibody (1/3000) (H-140, Santa Cruz Biotechnology, Dallas, TX, USA), for 1 h at room temperature and with the IRDye^®^ 800CW goat anti-rabbit IgG secondary antibody from LICOR for 1 h at room temperature, with three washes of 10 min each with PBS-T between incubations. Immunoblotting signals were analyzed by Odyssey Infrared Imager 9120 (LI-COR Biosciences, Lincoln, NE, USA). For small peptides, the primary antibody was anti-6xHis (1/5000) (Abcam, ab18184, mouse monoclonal) and IRDye^®^ 800CW goat anti-mouse IgG secondary antibody.

### 2.11. Fibrillarin Mutagenesis

Mutagenesis of the fibrillarin sequence was performed with the Thermo Scientific Phusion Site-Directed Mutagenesis Kit, using specific primers for each mutation: fwd 5′-TTTGGCGGGGGCGCGGGTCGAGGCGGA-3′, rev 5′-GCCCCCTCGGCCTCCACGAC-3′ and 5′-AGGTCGTGGAGCGGGAGGAGGTG-3′, rev 5′-CTAAAGCCTCCGCCTCGACC-3′ for R34A and R45A mutant, respectively.

### 2.12. RNA In Vitro Transcription

T7 RNA polymerase was used to in vitro transcribe on sense the U3 snoRNA cloned into the pGEM T-easy vector and SP6 RNA polymerase in antisense (New England BioLabs, Ipswich, MA, USA).

### 2.13. RNA Extraction from HeLa Human Cells

Total rRNA was extracted from the HeLa cell culture using a commercial kit, GenElute™ Mammalian Total RNA Miniprep (Sigma-Aldrich, St. Louis, MO, USA). 

### 2.14. In-Gel RNAse Activity Assay

Measurements were performed as zymography. Protein samples were separated in 15% SDS-PAGE gel. Resolving gel was supplied with 5 mg/mL of total extracted HeLa RNA prior to polymerization. After electrophoresis, the gel was washed for 10 min with buffer I (10 mM Tris-HCl, 20% isopropanol, pH 7.5) and incubated for 30 min in buffer II (10 mM Tris-HCl, pH 7.5) and buffer III (100 mM Tris-HCl, pH 7.5). The gel was resolved with 0.2% of toluidine blue and washed with water until the activity band was visible [42].

### 2.15. In Vitro RNA Activity Assay

Total RNA extracts from HeLa cells were mixed with HeLa expressed and purified SNAP-tag–fibrillarin complexes with or without the GAR domain; alternatively, recombinant fibrillarin was used alone in RNAse activity buffer (25 mM Tris-HCl pH8, 100 mM NaCl, 0.1 mM EDTA, 0.1 DTT, and glycerol 10%) + 0.8 to 1 U/µL of RNAse inhibitor (Thermo Fisher) and incubated for 45 min at 37 °C, then loaded in a 3% agarose gel. Phosphoinositides and PA were added to the final amount of 50 ng.

### 2.16. Fat Blot Assay

PIP strips were used for the fat blot assay (P-6001, Echelon^TM^, Santa Clara, CA, USA). The assay was made according to the manufacturer’s instructions with 0.4 μg of protein.

### 2.17. Fluorescence Recovery after Photobleaching

FRAP experiments were performed on a DeltaVision OMX Super-resolution microscope. HeLa cells were transiently transfected with pGFP-FIB and mutant constructs for the live cell analysis. DMEM media was replaced with warm phosphate-buffered saline medium (Dulbecco, Darmstadt, Germany) before the experiment. Dishes were placed in a temperature-controlled chamber supplemented with CO_2_. Images were obtained with a CMOs camera, using the 60X oil 1.42 objective with fluorescence free immersion oil 518F. For each photobleaching assay, one Cajal Body (CB) or nucleolus (Nco) was selected per cell and stated as the region of interest (ROI). The ROI was selected manually for immediate lase photobleaching (single point FRAP), then the fluorescence recovery was monitored, each experiment lasting 120 s, with one image acquired per second. Photobleaching removed ~99% of total fluorescence in both the nucleolus and CB. Image processing was performed with ImageJ® software for half-life (τ_1/2_) value extraction plus mobile and immobile fractions. R software was used to carry out quantitative and statistical analysis. Estimation of the diffusion coefficients (D) of wild-type and mutants was carried out as described in [43]. Each mutant was compared to wild-type (WT) under independent measurements; therefore the SD in the graph for WT is *n* = 50 for Nco and *n* = 50 for CB compared to each mutant *n* = 25 for Nco and CB, respectively.

### 2.18. Bioinformatic Analysis

Complete proteomes of 35 chordate species and *Saccharomyces cerevisiae* were used to retrieve GAR containing proteins. The complete list of species and their proteome files are listed in Appendix A. To identity GAR containing proteins, hidden Markov models (HMM) were constructed and calibrated from the alignment of 250 sequences retrieved by PSI-Blast [44] against the Refseq-protein database. The fibrillarin (*FIB*; AAP36189.1) and *hGAR1* (Q9NY12.1) protein sequences from *H. sapiens* were used as queries against the Chordata organism database (taxid:7711). The complete list of retrieved fibrillarin proteins is shown in Appendix A. An HMM model of the GAR domain was constructed from the alignment of the fibrillarin retrieved proteins and was subsequently used to detect other GAR domain-containing proteins in the 36 complete genomes. The list of retrieved proteins containing a GAR domain structure is given in Appendix A.

## 3. Results

### 3.1. Fibrillarin as a Ribonuclease

Unlike other proteins studied in our laboratory, we unexpectedly detected RNA degradation activity during the process of recombinant fibrillarin purification. Therefore, this effect was studied more deeply to clarify whether recombinant fibrillarin was the cause of degradation or whether a ribonuclease interacted with the fibrillarin preparations. We purified recombinant fibrillarin to near homogeneity (Figure 1a) after several chromatographic steps, as stated above.

Fibrillarin identity was verified Western blotting using an anti-Fib primary antibody (Figure 1b), and two bands were detected. The upper band corresponds to the expected size of full-length fibrillarin, whereas a lower molecular weight band may correspond to the degradation product of the main protein. An *in-gel* activity assay was used to confirm that the ribonuclease activity was from the purified fibrillarin (Figure 1c). One band coincided with the full-length of fibrillarin confirmed by Western blot (Figure 1b,c, Lane 2). In gel zymography, a small size band also exhibiting ribonuclease activity was noted (Figure 1c, Lane 2). This band corresponded with the potential fibrillarin degradation product detected by Western blot using an anti-Fib primary antibody (Figure 1b). Therefore, ribonuclease activity may be modular and present in a particular domain inside the fibrillarin sequence.

The interaction between fibrillarin and PIP2 has been reported [25]. Furthermore, according to the molecular characteristics for phosphoinositide binding [45,46], fibrillarin has the necessary amino acids (negatively charged and with aromatic rings) to bind phosphoinositides (Figure 1d). And indeed, in a phospholipid strip assay, fibrillarin interacted with phosphoinositides, but more strongly with the negatively charged PA (Figure 1e). From the homogeneous fibrillarin preparations, a dose-dependent ribonuclease activity in vitro assay was carried out by increasing the protein concentration from 1 to 6 ng with a constant amount of 2 µg of rRNA. The major 28/18S rRNA populations and their integrity used for the assay are depicted in Figure 1f Lane 2, and a DNA band co-purificated by TriZol® protocol is marked with arrowheads. rRNA degradation was directly proportional to the amount of fibrillarin added (Figure 1f, Lanes 2 to Lane 5); 28S rRNA was affected but not 5S rRNA (data not shown). The ribonuclease activity of fibrillarin was also found to be dependent on temperature, time, and display ion sensitivity (Appendix A).

Calcium ions have been detected in the crystal structure of human fibrillarin [47], and other enzymes with ribonuclease activity are either dependent on [48,49] or inhibited by calcium [50]. Our previous work showed that the AtFib2 from *A. thaliana* is activated with a small amount of calcium in ribonuclease activity assays, but no effect was observed in the same context with AtFib1 enzyme [51].

Ribonuclease activity was tested with PIP5 and PIP3 phospholipids with and without 1 mM Ca^+2^. The experiments were carried out with a reduced amount of fibrillarin (<1 ng) to observe the likely activation of the enzyme (Figure 1g,h). The addition of PI3P had an inhibitory effect for rRNA digestion (Figure 1g, Lane 3), whereas the addition of PI5P resulted in a dramatic increase of ribonuclease activation (Figure 1h, Lane 3). Of note, these phospholipids did not affect rRNA in the absence of fibrillarin. Thus, strikingly, PIP3 and PIP5 differently affected the ribonuclease activity of fibrillarin.

### 3.2. Ribonuclease Activity of the Recombinant Ribonucleoparticle Complex Involving Fibrillarin

Fibrillarin forms part of one of the main snoRNP complexes that modify rRNA during ribosomal RNA biogenesis; this complex guides the specific methylation site of about 100 residues in the rRNA. The complex consists of three other well-known proteins: Nop58, Nop56, 15.5K, and one snoRNA guide [52,53,54,55].

To examine the ability of fibrillarin to degrade rRNA in the RNP context, a construct [41] containing a truncated fibrillarin (from amino acid 82 to 321) with the GAR domain absent in the coding sequence was compared to full-length fibrillarin. The first purified complex with full-length fibrillarin was termed RNP, and the second purified complex with truncated fibrillarin was termed RNP ΔGAR. The two versions of the RNP complex were visualized on SDS-PAGE (Figure 2a); a shift in the fibrillarin corresponding size was observed.

The ribonuclease activity of RNP and RNP ⊗GAR complexes was tested in vitro using 2 µg of total RNA (Figure 2b); the complex with full-length fibrillarin degraded RNA, whereas RNP ⊗GAR showed no significant activity. The addition of 1 mM Ca^2+^ did not affect the ribonuclease activity of either complex (Figure 2c). Similar to the experiments with purified fibrillarin, the influence of phosphoinositides were evaluated at a final concentration of 5 ng, but no effect of PIP3 and PIP5 was observed (Appendix A). Alternatively, PIP2 inhibited the ribonuclease activity of the complex in a concentration-dependent manner (Figure 2d). We also checked by structured illumination microscopy (SIM) the localization of PIP2 and fibrillarin using a SNAP-tag–fibrillarin construct in a stable cell line, indicating that in the nucleoli, the majority of fibrillarin surrounded PIP2 in the DFC (Figure 2e) and that the ring-like structure involved a change of phase which lacks the lipidic environment outside the DFC. Fibrillarin ring structures are likely formed by the interaction of rRNA, non-coding RNA, and phospholipids. The ring-like structure has a diameter with an average of 488 nm (SD ± 7 nm), with a PIP2 core of 188 nm (SD ± 3.6 nm). An analysis of the ring-like structure and its comparison to other proteins can be found in Appendix A to Appendix A. In conclusion, the ability of fibrillarin to interact with several phospholipids may provide different functions in different phases from FC to DFC.

### 3.3. Fibrillarin Specificity Ribonuclease Activity in Complex with RNA Guide

The specificity of the SNAP-tag–fibrillarin complex as a ribonuclease was tested against U3 snoRNA, one of several RNA processing guidelines for rRNA that interacts with fibrillarin (Appendix A). Fibrillarin did not degrade U3 snoRNA, but interacted with it, resulting in retardation, as confirmed by GMSA (Figure 3a). The addition of PA dissociated the complex from U3 snoRNA (Figure 3a), whereas PIP2, under the same conditions, showed a minor alteration in the complex migration on gel (Figure 3a, Lanes 3 and 5).

Fibrillarin was not able to degrade rRNA when pre-incubated with snoRNA U3 for 15 minutes, prior to rRNA addition, suggesting that U3 snoRNA blocked the ribonuclease activity of fibrillarin (Figure 3b, Lane 7). When fibrillarin was incubated for 15 minutes without U3 snoRNA followed by the addition of rRNA, degradation was observed as expected. HeLa expressed and purified SNAP-tag–fibrillarin complex also showed ribonuclease activity (Figure 3c). An in-gel activity assay of the HeLa purified fibrillarin complex had 5 bands with ribonuclease activity corresponding to four bands revealed by Western blotting with anti-fibrillarin antibody (Figure 3c; marked with arrowheads). Fibrillarin degradation is likely to have occurred during the complex purification, as fibrillarin is prone to degradation even in the presence of protease inhibitors. The fifth band (asterisk) showed as-yet-undetermined ribonuclease activity. Fibrillarin complex from HeLa cells was unable to cut snoRNA U3 (Figure 3d). The subcellular distribution of PA was determined using a PA sensor described previously [39]. In HeLa cells during interphase, on one side, fibrillarin is concentrated in the nucleolus and, on the other side, the majority of a PA sensor was detected in perinuclear structures reminiscent of the Golgi apparatus with very little overlap of the signals (Figure 3a). Upon nuclear membrane rupture during mitosis, fibrillarin is abundantly relocated to the cytoplasm, where it colocalized with the PA sensor, as shown in Figure 3e. Note that during mitosis, the vesicular distribution of the PA sensor completely disappears. It is thus possible that fibrillarin interacts with PA in the cytoplasm during mitosis. The presence of PA inhibited the in vitro activity of the full-length fibrillarin containing the recombinant complex. This inhibition was not dependent on the size of the acyl chains of PA (Figure 3f, Lanes 4 and 5). A small change in the digestion pattern was, however, observed with low amounts of PIP2 (Figure 3f, Lane 3).

### 3.4. GAR Domain, Modular in Fibrillarin

Next, we evaluated the ability of the GAR domain from the recombinant complex to act as an active domain. The N-terminal region (GB: 1-134 aa) of fibrillarin includes the GAR domain and a sequence of amino acids 58 to 136 (the BCO) that has no defined function [10]. The GB (GAR and BCO regions) domain was cut into several sections, as depicted in Figure 4a. Constructs fused to GST and 6xHis were expressed and purified by a double-step purification and detected by a specific antibody (Figure 4b,c). The GAR domain exhibited ribonuclease activity, whereas the BCO did not show any activity (Figure 4d, Lane 5). Further deletions of amino acids 1 to 20 and 13 to 68 of the GAR domain abolished activity (Figure 4d, Lanes 4 and 6, respectively).

An in-gel ribonuclease activity assay showed that the GAR domain fused with GST degraded RNA, and its activity increased when GST was cleaved (Figure 4e, Lanes 3 and 5). Increasing concentrations of purified GAR domain caused significant degradation of rRNA at the highest amount tested. Interestingly, 28S rRNA was the first target for the degradation using this domain (Figure 4f).

### 3.5. Mutation of the GAR Domain

From sequence analysis of the GAR domain, the conserved arginine in position 34 and 45 was chosen to be substituted with alanine (Figure 5a). Presumably, the alteration of amino acids causes a conformational change that better exposes the GAR domain in the R45A mutant, but not in R34A. The amount of all pure recombinant fibrillarins was normalized for ribonuclease activity or phospholipid-binding assays (Figure 5b,c). Surprisingly, R34A and R45A fibrillarin mutants lost the ability to interact only with the PIP2, PA, and phosphatidylserine (PS), while interaction with PI3P and PI5P remained. R34A mutant also interacted weakly with phosphatidylcholine (PC) and sphingosine-1-phosphate (S1P) (Figure 5c). In conclusion, both mutations affected interaction with several anionic phospholipids, as shown by fat blot analysis, in agreement with the model where the positive arginine charges are essential for anionic phospholipid binding. WT fibrillarin and the R34A fibrillarin mutant degraded the rRNA 28S to the same degree, but the R45A fibrillarin mutant was found to be more active and degraded up to 80% of 28S rRNA (Figure 5d).

WT fibrillarin, mutant R34A, or R45A genes fused to GFP showed the previously reported intranuclear localization (enrichment in nucleoli and Cajal bodies) [1,2], but different intranuclear behavior in transiently transfected in HeLa cells. FRAP analysis showed that fibrillarin mutants had a 40% lower diffusion coefficient in Cajal bodies (CBs) showing statistical difference in *p*-values CB (WT - R34) *p* = 0.0056 (**), and (WT - R45) *p* = 0.0027 (**). The mutants showed minor reduction in mobility in the nucleus (Nco) with *p*-values for Nco (WT - R34) *p* = 0.0450 (*) and for (WT - R45) *p* = 0.4302 (ns). Nucleolar R34A and R45A fibrillarin mutants showed a significant diminution in dynamic speed compared to WT, as reflected in increased half-lives. Compared to WT fibrillarin, R45A had a 50% reduction in nucleoli size, of which the intranuclear localization is shown in Figure 5e. In CBs, there was also an increase in the half-lives of R45A and R34A versus WT fibrillarin, reflected in smaller diffusion coefficient values observed as the longer residence time of the mutants within the nucleolus and CBs in comparison with WT (Figure 5f).

### 3.6. Ribonuclease Activity of GAR-Like Domains Containing Proteins

The phylogenetic tree constructed from the alignment of the GAR domain of the retrieved fibrillarin proteins was grouped into 3 major clades (Figure 6a; protocol and data in Appendix A). The basal branch contained the GAR domain of NOP1 protein from *S. cerevisiae*, the most ancestral species analyzed, and sequences of non-mammalian species. Group A was divided into 2 subclades (A1 and A2) and contained nearly all vertebrate species with single fibrillarin sequences. Group A contained mammalian sequences only, as compared to the basal group. Subgroup A1 contained GAR sequences from non-primates except for the Gorilla GAR sequence. Species with a second fibrillarin sequence were clustered in group B and exhibited a distinctive GAR pattern (Appendix A).

Using the HMM model constructed from the GAR domain of fibrillarin proteins, five different protein classes were retrieved: the fibrillarin proteins (fibrillarin domain), LSM14-A proteins (containing LSM14 (Scd6-like Sm domain and FDF domain), a predicted autoimmune regulator in armadillo (*Dasypus novemcinctus*, containing HSR domain and PHD-finger), a predicted nucleolin isoform in rabbit (*Oryctolagus cuniculus*, containing 4 repeats of RRM1 (RNA recognition motif, RRM, RBD, or RNP domain), a nucleolar RNA helicase 2 in Zebrafish (*Danio rerio*, containing a DEAD (DEAD/DEAH box helicase), and a Helicase C (Helicase conserved C-terminal domain) and a GUCT (NUC152) domain). All 5 retrieved proteins contain domain regions with repeats of glycine-arginine amino acids known as RGG boxes, which are present in several proteins that participate in transcription, RNA binding and splicing and protein interactions [56,57,58]. A representation of the GAR domain location in each of the 5 retrieved proteins is depicted in Figure 6b. HMM for RGG-box 1 and RGG-box 2 repeats from GAR1 proteins were constructed, and other GAR1 proteins were detected in the analyzed genomes (Appendix A). LMS14-A was detected in 20 different species and contained 2 different R/G boxes, one before the FDF domain (a 20 aa R/G box I) and the other after the FDF domain (a 44 aa R/G box II). A multiple-sequence alignment of 23 LSM14-A proteins was performed and visualized with the Boxshade tool available at https://embnet.vital-it.ch/software/BOX_doc.html. The structure of the R/G rich region in such proteins followed the characteristic motif of the GAR domain (Appendix A). Considering that the GAR domain is absent in Archaebacteria fibrillarins and that it acts as a modular domain in other proteins, the published genomes for similar sequences were revised. 

The Barley Stripe Mosaic Virus (BSMV) Triple Gene Block1 (TGB1) is a 58 kDa movement protein that possesses RNA-binding, RNA helicase, and ATPase activities [59,60,61]. The BSMV TGB1 protein contains a nuclear localization signal (NLS) and a nucleolar localization signal (NoLS) between 227–238 and 95–104 amino acids, respectively. BSMV TGB1 protein interacts with the GAR domain of the nucleolar fibrillarin (Fib2) from *Nicotiana benthamiana* [62]. Other examples correspond to the ORF3 viral protein that also interacts with the fibrillarin GAR domain [63]. Some NLS and NoLS contain intrinsically disordered regions [64,65] that help direct viral proteins to the nucleolus during viral infection. Therefore, the potential for TGB1 to degrade RNA was evaluated.

Cloning and expression in *E. coli* were performed for the GAR domains from Lsm14, full-length GAR1, and the viral protein TGB1, before IMAC purification (Figure 6c), and their ribonuclease activity was compared in vitro with that of the GAR domain of fibrillarin. As shown in Figure 6d, the GAR domains from fibrillarin, LMS14, and GAR1 presented different degrees of activity as compared to the activity of RNAse A (Lane 3), which degraded total rRNA. In contrast, GARs from fibrillarin (Lane 4), Lsm14 (Lane 5), and GAR1 (Lane 6) presented a different pattern of degradation and null activity, as in the case of TGB1. Thus, not all the intrinsically disordered regions presented activities related to rRNA processing (Lane 7). We hypothesized that glycine–arginine repeat domains raised during evolution as a part of the lipid complexity and functional compartmentalization are required in the nuclear environment.

## 4. Discussion

Fibrillarin is an essential protein whose function and structure are highly conserved in all eukaryotic organisms [10,66]. It functions as a methyltransferase involved in the processing of rRNA [67] and the methylation of histone H2A in RNA Pol I promoters [12,13]. Here we describe its activity as a ribonuclease even in the absence of a complex with Nop56, Nop58, and 15.5K proteins. Ribosomal biogenesis requires strict control and tight regulation to maintain a high order of rRNA production [68,69]. The 47S pre-rRNA transcribed binds co-transcriptionally to fibrillarin in the periphery of the FC/DFC in its 5’ end, and the self-assembly of fibrillarin in its GAR domain facilitates a directional sense of the pre-rRNA processing [70], forming specific clusters of RNA-protein complexes that mediate subsequent modifications of pre-rRNA after transcription. The ring-like structures are highly ordered and may involve a phase separation for fibrillarin activity. RNA pol I, together with UBF, are localized in the center with the majority of the nucleolar PIP2 pool. The majority of fibrillarin is found in this surrounding area, as seen in Figure 2e. The particular ring structures are similar in size and shape to those previously identified by labeling SLERT. The RNA SLERT is known to associate and regulate pre-rRNA transcription and rRNA production, leading to increased tumorigenesis [71]. 

The initial possibility for ribonuclease activity of fibrillarin was suggested by Kass et al. [18], who determined that a ribonucleoprotein complex containing fibrillarin and snoRNA U3 was involved in specific ribonuclease activity, using nuclear extracts from mouse cells. Furthermore, an antibody against fibrillarin decreased rRNA digestion significantly [18]. Tollervey and collaborators [67], using NOP1 (yeast fibrillarin), showed that its depletion reduced the level of mature rRNA but increased the amount of uncut premature rRNA [72]. In cruciferous plants, Vasquez-Saez et al. [22] isolated the ribonucleoprotein complex nuclear factor D consisting of 30 proteins, including fibrillarin, nucleolin, and the snoRNAs U3 and U14. The complex can interact with rDNA and cut into the P site, downstream from the A1, A2, A3, and B sequences located in the 5’ ETS of pre-rRNA. However, the ribonuclease activity was not associated with any particular protein. In *C. elegans*, fibrillarin is a key regulator of pathogen resistance during bacterial pathogen infection [73]. In vitro and biomolecular fluorescence complementation assays [74] showed in *A. thaliana* that fibrillarin (FIB2 gene) can interact with the ELF18-INDUCED LONG NONCODING RNA 1 (ELENA1) lncRNA and directly with MED19a in the nucleoplasm and the nucleolus, demonstrating that the FIB2 gene is a negative transcriptional regulator of immune responsive genes like PR1. We found here that the GAR domain of fibrillarin is responsible for its activity as a ribonuclease. The GAR domain is present in several nucleolar proteins, i.e., nucleolin [75], NSR1 [76], SSB1 [77], and GAR1 protein [78], which have RNA recognition motifs like fibrillarin, directly interact with rRNA, and are involved in its processing [78,79]. In fibrillarin, the GAR domain is needed to target nucleoli [80] and for interaction with nuclear phase viruses [63].

There is an increasing suggestion about how LLPSs are regulated in relation to the post-translational modifications (PTMs) on the disordered domains of different nucleolar proteins that promote phase separation behavior. Cross-talk between PTMs such as phosphorylation and methylation regulates and affects interactions between protein partners or other molecules. NOP1, the fibrillarin yeast counterpart, can be methylated at six of nine SRGG motifs, which belong to the GAR domain, and this arginine methylation increases the ability of NOP1 to interact with the major RNP proteins Nop56 and Nop58, also regulating the localization of NOP1 during ribosomal processing [24]. The mutations we carried on two arginines in fibrillarin led to an alteration in nucleolar size and dynamics associated with an alteration to phosphoinositide binding.

Furthermore, we established the interaction of various phospholipids with fibrillarin, including phosphoinositides and PA. Within the cell nucleus, phosphoinositides are essential cofactors for various processes ranging from transcription regulation and differentiation to cell cycle control [28,81]. Previously we described that PIP2 interacts with fibrillarin affecting its RNA binding mobility in native PAGE [25]. Here we also show that fibrillarin also interacts with PA. The general ribonuclease activity decreases upon interaction with PA and releases fibrillarin from the interaction with the guide snoRNA U3. We suggest that during mitosis, the interactions between PA and fibrillarin may prevent unspecific degradation of RNA and allow different complex formation.

Negatively charged phospholipid-binding domains usually require positive charges. A clear example is the γ-core, where the arginine has been mutated in a signature similar to that of the GAR domain of fibrillarin. Such mutations show similar alteration of phosphoinositide binding [82] as in our mutants. The mutation made in the GAR domain of R45A has a more significant effect on the activity of fibrillarin as a ribonuclease. Both mutants (R34A and R45A) modify the interaction with negatively charged phospholipids compared to WT fibrillarin. Phospholipids affect binding to proteins and can promote protein-protein interaction. An example is the PDZ domain (PSD-95/Discs large/ZO-1) of syntenin-2 [83]. This domain interacts with PIP2, regulating the nuclear organization of some proteins bearing this domain. The mutants for this PDZ domain (in positions Lys113, Lys167, Lys197, and Lys244) result in a loss of binding with PIP2 and functional alteration [83]. The nucleolar size changes in mutants may result from changes in protein interactions, as indicated by our FRAP experiments. Since fibrillarin interacts with at least 235 proteins [10], the mutations may lead to a reduction in size due to improper complex formation, resulting in a partial compromise in nucleoli architecture.

Ribosome biogenesis is an essential cellular process that consumes cell energy [84]. During interphase, where a high degree of ribosome synthesis is required, fibrillarin is mostly deacetylated by the NAD+ dependent deacetylase SIRT7, increasing H2AQ104 methylation levels and ensuring rRNA synthesis by Pol I complex activity. Here we show that fibrillarin also acts as a ribonuclease regulated by phosphoinositides and PA. It is favorable for the cell to inhibit a ribonuclease like fibrillarin during metaphase, and PA may provide such a mechanism. The PA found in membranes in the cytosol may interact with fibrillarin once the nuclear membrane is broken during metaphase. However, clearly more research in the subject is needed before this could be concluded. Fibrillarin ribonuclease activity may be directed by some guide RNA, like snoRNA U3, together with a phase change for functional localization. We used a complete processed rRNA without the external and internal transcription spaces, in which U3 was involved to guide the cleavages in at least three different positions [18]. This is the reason why the complex fibrillarin-U3 snoRNA was not able to degrade rRNA. Yeast mutants of NOP1 show different phenotypes that affect ribosome function and rRNA biogenesis at different stages, including a lack of cleavage of the pre-rRNA. Here we show for the first time that human fibrillarin has a second activity that is essential for such processes [10,51].

## Figures and Tables

**Figure 1 cells-09-01143-f001:**
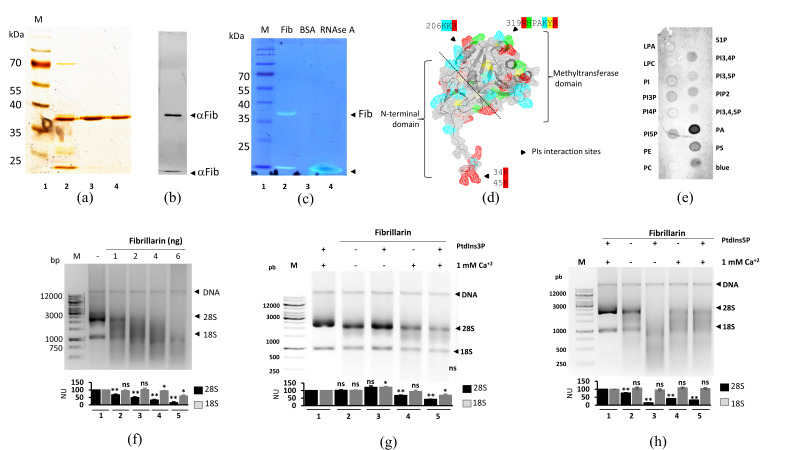
Fibrillarin acts as a ribonuclease. (**a**) Silver stain of purified fibrillarin. M: molecular weight marker. 2, 3, and 4: different fibrillarin elution profiles. Only elution 2 and 3 (major quality of protein purification) were used for further analysis. (**b**) Fibrillarin Western blot revealed two bands marked with arrow-heads that were recognized by specific fibrillarin antibody. (**c**) In-gel fibrillarin RNA assay shows two signals marked with arrowheads corresponding to fibrillarin (Lane 2). BSA and RNAse A were used as negative and positive controls, respectively (Lanes 3 and 4). (**d**) Computational inference of possible binding sites for phosphoinositides, one residing in the GAR domain and two in the globular C-terminal domain, are marked with arrows. The fibrillarin sequence with characteristic amino acids for phosphoinositide-binding is noted. Three to five amino acids are necessary for phophoinositide-binding [45,46]. These amino acids should be positively charged with at least one having an aromatic ring. Red: arginine, blue: lysine, green: histidine, yellow: tyrosine. (**e**) Fibrillarin fat blot assay shows that fibrillarin interacts modestly with several phosphoinositides and strongly with PA. Appendix A depicts the quantification signals made by ImageJ software. (**f**) Total RNA in vitro assay was performed while increasing the concentration of protein (from 2 to 8 ng) added to a constant concentration of rRNA (2 μg). Degradation of RNA is directly proportional to the amount of fibrillarin. Copurified DNA from TriZol extraction is indicated by head arrows; the rRNA populations, i.e., 28S and 18S, are indicated. Influence of PI3P (**g**) and PIP5 (**h**) on human fibrillarin in vitro ribonuclease assay. (**f**–**h**) Quantification of 28S and 18S signals were made by ImageJ software and represented in normalized units (NU), statistical significance was determined by *t*-test (* *p*-value < 0.05; ** *p*-value < 0.01; *** *p*-value < 0.001, ns= not significant *p*-value > 0.05) and plots are indicated below the gels from *n* = 3.

**Figure 2 cells-09-01143-f002:**
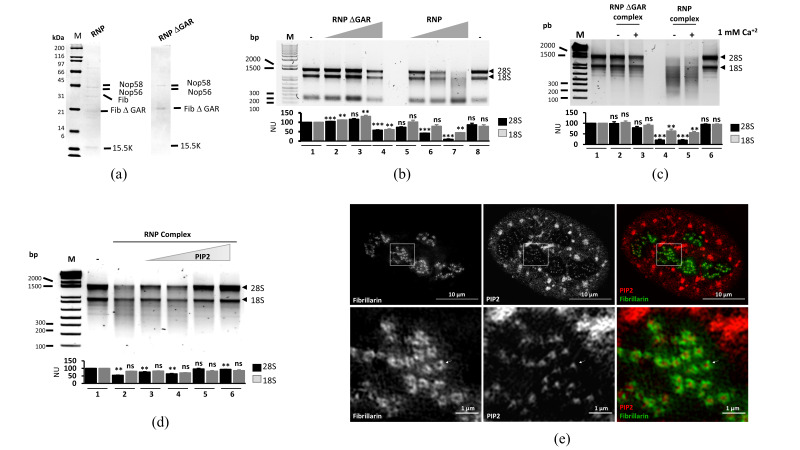
RNA activity of the recombinant ribonucleoprotein complex containing a truncated (RNP ΔGAR) and full-length complex (RNP). (**a**) IMAC affinity purification under native conditions of RNP ΔGAR and RNP complexes. The sequence coding GAR domain was inserted into the truncated fibrillarin–RNP complex, using the EMP–PCR strategy. A shift in the band size corresponding to fibrillarin is shown. The Nop58, Nop56, and 15.5K proteins are copurified as expected and labeled in the figures in both RNP purifications (**b**) Comparative activity of complexes with or without GAR domains. No activity was detected using increasing amounts of native RNP ΔGAR, while RNA degradation was observed with full-length fibrillarin in the complex. (**c**) Influence of calcium at 1 mM on the RNA activity. No significant modification of activity was observed between the complexes. (**d**) In vitro RNA assay using increasing amounts of PIP2 lipid. Increasing amounts of phospholipid from 1 to 100 ng inhibited the of RNA activity. (**b**–**d**) Quantification of 28S and 18S signals were made by ImageJ software and statistical significance was determined by *t*-test (* *p*-value < 0.05; ** *p*-value < 0.01; *** *p*-value < 0.001, ns = not significant *p*-value > 0.05) and plots are indicated below the gels from *n* = 3. (**e**) Colocalization of SNAP-tag–fibrillarin and immuno-labeled PIP2 (left panel) in HeLa cell nucleus. White inset is magnified in the right panel; the red signal corresponds to PIP2 and the green one to SNAP-tag–fibrillarin.

**Figure 3 cells-09-01143-f003:**
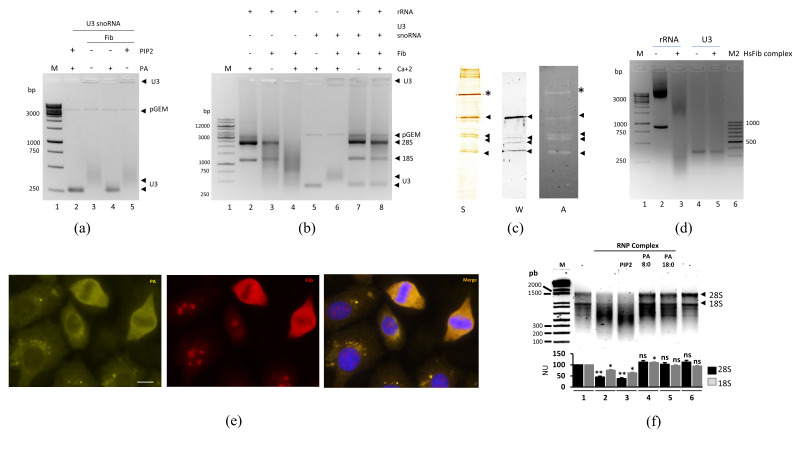
RNA specific RNA activity by fibrillarin with snoRNA U3. (**a**) Gel shift of snoRNA U3 with WT fibrillarin. All incubations and procedures carried out at 4 °C, PIP2 and PA were added as indicated. Plasmid control pGEM was used to normalize each lane. (**b**) RNA activity of fibrillarin with or without guide snoRNA U3 added. Reactions were carried out at 37 °C. (**c**) In-gel RNA activity of native fibrillarin RNP complex, labeled with silver staining (S), revealed by Western blot (W) and tested by activity assay (A). Arrowheads indicate fibrillarin and potential degradation products, and an asterisk indicates an undetermined ribonuclease activity (**d**) In vitro RNA activity of HeLa expressed and purified SNAP-tag–fibrillarin complex against total RNA and snoRNA U3. (**e**) Immunofluorescence showing the colocalization of PA and fibrillarin. The signal corresponds as follows: Blue–DAPI (nucleus signal), Yellow–PA sensor, and RED–fibrillarin. (**f**) In vitro RNA assay using PA. An inhibition effect was observed using PA containing 8 or 18 carbons and no saturation in the lateral chains (8:0 or 18:0), whereas little effect on the activity was observed with PIP2. Quantification of 28S and 18S signals were made by ImageJ software and statistical significance was determined by *t*-test (* *p*-value < 0.05; ** *p*-value < 0.01; *** *p*-value < 0.001, ns = not significant *p*-value > 0.05) and plots are indicated below the gels from *n* = 3.

**Figure 4 cells-09-01143-f004:**
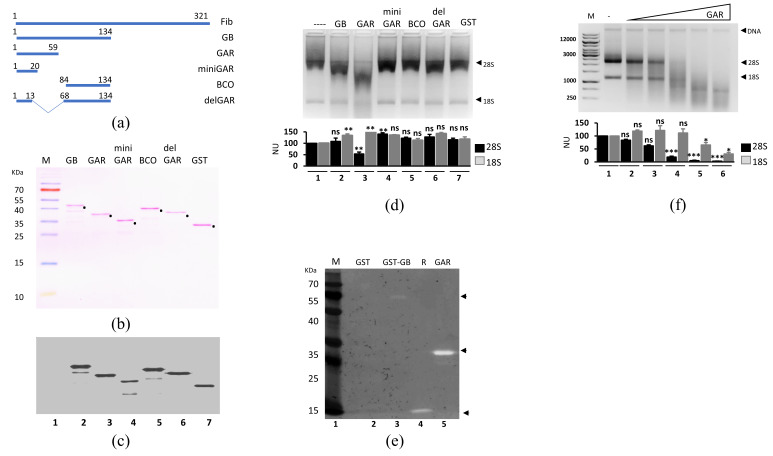
The GAR domain of fibrillarin is a modular domain with RNA activity. (**a**) Schematic representation of small peptides expressed from GB. (**b**) Ponceau staining of the HsGB small peptides expressed. All peptides were fused to the GST and 6 His tags. The band corresponding to the expected peptide is shown by a dot. **(c)** Anti-6xHis antibody was used to identify each peptide by Western blot. (**d**) RNA assay from the purified peptides. Under the same conditions, the GST-GAR domain has higher activity than the GST-GB domain (Lanes 2 and 3). The other peptides (miniGAR, BCO, and delGAR) have no activity. GST was used as a negative control. (**e**) In-gel RNAse assay of fibrillarin GB domain (amino acids 1 to 134). Degradation of RNA is higher when GST was cleaved from the GAR domain (Lanes 3 and 5). GST and RNAse A (R) were used as negative and positive controls, respectively. (**f**) RNA activity assay of the GAR domain. rRNA degradation is directly proportional to the amount of the GAR domain. (**d**,**f**) Quantification of 28S and 18S signals were made by ImageJ software and statistical significance was determined by *t*-test (* *p*-value < 0.05; ** *p*-value < 0.01; *** *p*-value < 0.001, ns = not significant *p*-value > 0.05) and plots are indicated below the gels from *n* = 3.

**Figure 5 cells-09-01143-f005:**
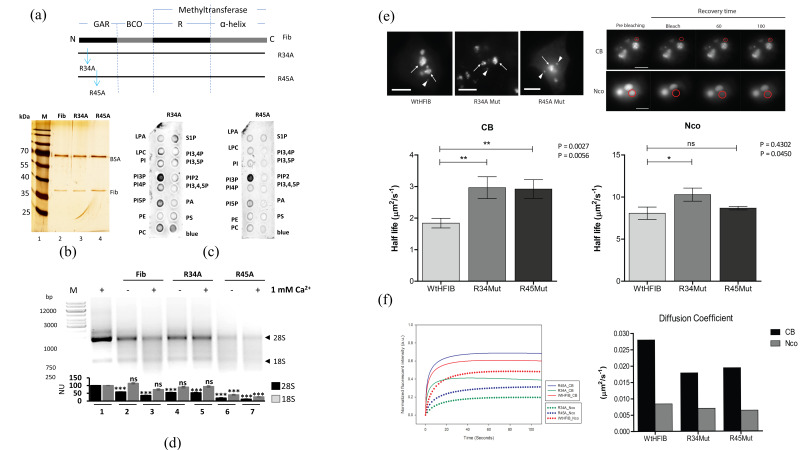
Activity of GAR domain mutants. (**a**) Schematic representation of fibrillarin domains GAR, BCO, and RNA binding alpha-helix domain. Blue arrows represent mutations R34A and R45A in the fibrillarin GAR domain. (**b**) Silver stain normalization of WT, R34A, and R45A fibrillarin. (**c**) Fat blot assay for fibrillarin mutants R34A and R45A. Appendix A depicts the quantification signals made by ImageJ software. (**d**) RNA activity of fibrillarin mutants R34A and R45A. Quantification of 28S and 18S signals were made by ImageJ software and statistical significance was determined by *t*-test (* *p*-value < 0.05; ** *p*-value < 0.01; *** *p*-value < 0.001, ns = not significant *p*-value > 0.05) and plots are indicated below the gels from *n* = 3. (**e**) Transiently transfected HeLa cells expressing GFP-WT-fibrillarin, R34A, or R45A mutant coupled to GFP. In the upper left panel, representative images of the conventional localization of human fibrillarin in nucleoli (arrows) and Cajal bodies (arrowheads) are shown for WT and mutants. Intranuclear localization for WT fibrillarin and its mutants was observed in live cells. On the upper right panel, one representative photobleaching experiment is shown for a Cajal body and, for a nucleolus, red circles delimit the ROI, which was subsequently photobleached. The lower graphs plot the half-life coefficients obtained from 200 independent FRAP experiments comparing WT and mutants fibrillarins in CBs and NCOs; error bars correspond to SEM. (**f**) The left graph represents the normalized dynamics from 200 independent photobleaching events for WT and mutant fibrillarins in CBs and NCOs; the curves in the graphs represent the normalized values of the mean (*n* = 200) for each condition for each time point. The bar graph on the left shows a resume of the single diffusion coefficient values obtained from the previous analysis.

**Figure 6 cells-09-01143-f006:**
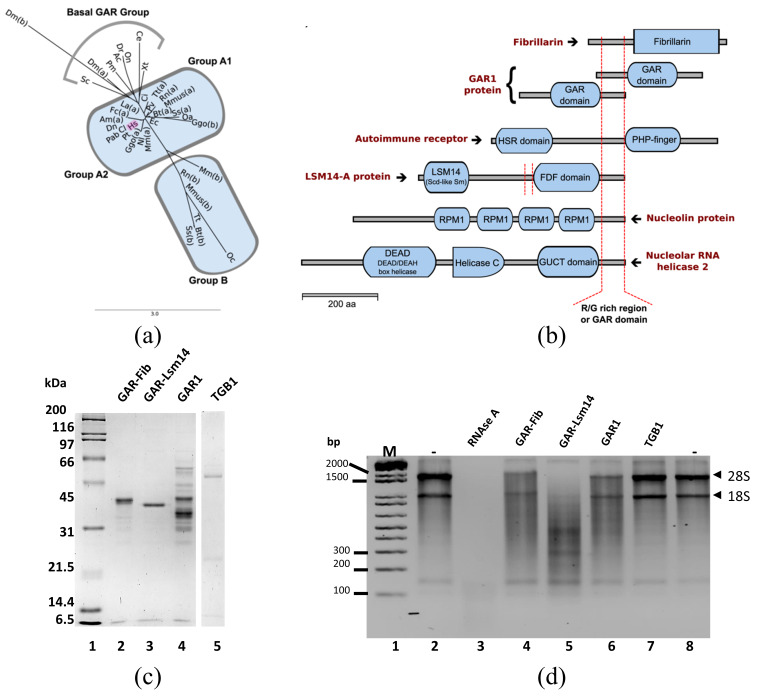
RNA activity of GAR domains from different RNA-binding proteins. (**a**) Bioinformatic analysis of R/G rich region (or GAR domain) of retrieved proteins from 36 complete genomes. The phylogenetic tree of GAR domains of fibrillarin proteins was retrieved from chordates genome data. GAR domains were clustered in three major clades: A, B, and the basal group. Clade A was divided into A1 and A2 subgroups. (**b**) Schematic representation of different modular proteins shows the presence of GAR domains in N- or C-terminal regions. Five proteins contained GAR domains similar to fibrillarin: GAR1 protein, an autoimmune receptor, Lsm14, Nucleolin, and a nucleolar RNA helicase 2. (**c**) IMAC affinity purification of GAR domains from fibrillarin, Lsm14, GAR1, and viral protein TGB1 with a disordered domain similar to GAR. (**d**) In vitro RNA assay of four GAR domains expressed and purified from *E. coli.* TGB1 and GAR1 (Lanes 6 and 7) showed no activity against total RNA. RNAse A diluted 1:20,000 was used as control.

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
