# Peer review of "Fibrillarin Ribonuclease Activity is Dependent on the GAR Domain and Modulated by Phospholipids"

_cells, 2020, doi:10.3390/cells9051143_

Round 1
Reviewer 1 Report
The paper by Guillen-Chable et al, describes the activity of Fibrillarin and shows that this protein has novel function as a ribonculease. They also showed that this ribonuclease activity is modulated by phospholipds. They when further to mapped the region in the Fibrillarin that is reponsible for the ribonuclease activty being in the GAR domain. In addition, they did mutagensis analysis to pipoint the residues responsible for phospholipd binding. Finaly, they did phylogenetic analysis and identify anallogue proteins with similar GAR domains. They confrmed this by expressing and testing the function of a few of these proteins.
I find the work well planned and executed and the findings well reported.
On the other hand, I think that the authors need to do a bit more on the following to make their manuscript readily accepted for publiction
-
Consistency in naming and the abbreviation throughout the mamuscript for example, in the materials and method, the authurs stated “63°C, 30 seg), and extension (72°C, 15 seg)” am not sure what “seg” is for? Normally you will write seconds or sec. This is found throughput the text. Line 91 they started a sentence as “The media from the disk disacarded...” what disk is this? Line 245 “DMSM” what does this stand for? Line 271 “we incidentally found that showed RNA....”, should be revised. The use of RNP and fibrillarin is not consistent throughout the text and are interchangaeble and confusing. The sentences between lines 357-360 should be revised. Could not follow. Contains repetition not very clear meaning.
-
While the influence of phospholipds was clearly shown, it will have been nice to put this in a bigger context by relating this to the function of the cell and decribing mechanistically how this work. For example, how the different domains interact and how the interaction of the specfic lipids regulate this by interacting with the GAR domain.
Author Response
Consistency in naming and the abbreviation throughout the manuscript for example, in the materials and method, the authors stated “63°C, 30 seg), and extension (72°C, 15 seg)” am not sure what “seg” is for? Normally you will write seconds or sec.
We apologize about the improper abbreviation used for the word “seconds”, the correcting changes were made at the 2.4 section (line 188), 2.5 section (lines 208, 209, 217 and 218). We checked for nomenclature consistency throughout the text.
Line 91 they started a sentence as “The media from the disk disacarded...” what disk is this?
We apologize for using the lab nomenclature for the Petri dishes and changed the text by “The media from the HeLa cell cultures was discarded and the cell monolayer washed …”, in Line 238.
Line 245 “DMSM” what does this stand for?
We apologize for the typo, which was corrected by DMEM (Dulbecco’s Modified Eagle’s Medium), in Line 294
Line 271 “we incidentally found that showed RNA....”, should be revised.
The previous version starting as: “During purification of recombinant fibrillarin we incidentally found RNA degradation activity…” was changed in lines 321-324 by “Unlike for other proteins studied in our laboratory, we unexpectedly detected RNA degradation activity during the process of recombinant fibrillarin purification. Therefore, this effect was studied more deeply to clarify whether recombinant fibrillarin was the cause of degradation or whether a ribonuclease interacted with the fibrillarin preparations”
This is found throughput the text. The use of RNP and fibrillarin is not consistent throughout the text and are interchangeable and confusing.
In order to make a more consistent use of RNP, the abbreviation “RNP” is stated only for the ribonucleoprotein complexes, which includes fibrillarin, Nop58, Nop56 and 15.5K proteins.
The sentences between lines 357-360 should be revised. Could not follow. Contains repetition not very clear meaning.
We apologize for this confusing sentence and it was revised lines 419-421 of the new version of the manuscript by “The ribonuclease activity of RNP and RNP DGAR complexes was tested in vitro using 2 µg of total RNA (Figure 2b); the complex with full-length fibrillarin degraded RNA, whereas RNP DGAR showed no significant activity”.
While the influence of phospholipds was clearly shown, it will have been nice to put this in a bigger context by relating this to the function of the cell and describing mechanistically how this work. For example, how the different domains interact and how the interaction of the specfic lipids regulate this by interacting with the GAR domain.
As kindly suggested by Reviewer 1 we now discuss the influence of phosphoinositides on the localization and phase separation processes into a bigger context, as cited in [28,29]. The recent hypothesis around the involvement of phosphoinositides that regulate important processes is getting more attention because several proteins involved in nuclear function possessing intrinsically disordered regions that bind RNA molecules also interact with phosphoinositides. We focused our discussion on PIP2 as there is a wealth of data on how this phospholipid is localized in particular and distinguishable functional structures in the cell nucleus.
Reviewer 2 Report
The manuscript by Guillen-Chable et al. provide evidence to suggest that the highly conserved and essential nucleolar methlytransferase fibrillarin, which is known to methylate ribosomal RNA (rRNA) and the histone protein H2B, also possesses ribonuclease activity. The authors show that the ribonuclease activity of fibrillarin is sensitive to the presence of phophoinositides and phospatidic acid and insensitive to treatment with ribonuclease inhibitors. Specifically, they show that the presence of phosphatidic acid inhibits the ribonuclease activity of fibrillarin and dissociates it from the guide snoRNA U3. The authors map the ribonuclease activity of fibrillarin to its G/R rich (GAR) domain, which they show is conserved in a small group of RNA-binding proteins. Finally, the authors provide a phylogenic analysis that suggests that it arose in evolution during the transition from Archaea to Eukaryotes. Overall, I feel that this manuscript is not acceptable for publication in its current form due to several major and minor issues, which I will outline below.
Major Issues
- The majority of the results presented in this work are not quantified, which makes it difficult to assess their rigor and reproducibility. Moreover, there are no statistical tests performed to evaluate the significance of the authors’ results. This is simply not acceptable in this day and age.
- The Materials and Methods section is rather disorganized and does not clearly describe how many of the experiments presented in this manuscript were performed. Several examples are provided below.
- In section 2.8, the authors describe the use of the “in vitro labeling with SNAP-Biotin strategy” for labeling their native fibrillarin. However, the authors do not describe having ever tagged fibrillarin with the SNAP-tag. They go on to state that they used Biotin-streptavidin beads to purify fibrillarin from cell lysates without identifying which cells were used. Thus, can the authors really say that they are purifying “native fibrillarin”, as the title of section 2.8 suggests?
- The Materials and Methods does not mention anything about the use of a fluorescent SNAP-tag-binding ligand. Thus, I am confused as to what exactly the authors did to generate the image show in Figure 2E.
- In section 2.16, the authors do not explain how they were able to select a region of interest for performing their FRAP experiments. Since the DeltaVision OMX is not a confocal microscope, this information is necessary for the reader to be able to critically assess the FRAP experiments presented in Figures 5E-F. Also, what do the arrows and arrowheads signify in Figure 5E?
- How many cells were the 200 independent FRP measurements of WT and mutant fibrillarins performed in?
- How were the “RNPs with or without the GAR domain sequence” assembled? While section 2.7 of the Materials and Methods describes the expression and purification of these RNPs, the information is scant at best.
- How did the authors fluorescently label PIP2 in the SIM image shown in Figure 2E? The probe does not seem to be described anywhere in the manuscript.
- What is a “RGG-box1” model?
- What is a “Boxshade analysis”?
- Is the fibrillarin that the authors purified in this work functional? In other words, does the purified fibrillarin perform its established job as an rRNA or H2A methlytransferase?
- The authors show a plot of the results of their FRAP experiments in Figure 5F without providing any images to show how or where the photobleaching was performed in cells expressing the indicated fibrillarin-EGFP constructs. These images are necessary for the reader to critically evaluate, not to mention believe, these experiments.
- The authors conclude from their single SIM image shown in Figure 2E that in nucleoli “the fibrillarin surrounded PIP2 in the DFC and the ring-like structure involved a change of phase which outside the DFC lacks the lipidic environment”. However, I cannot see how they are able to conclude anything about the phase transition of fibrillian from a single static image.
- I would caution the authors from making drawing too many conclusions from the images presented in Figure 3E regarding the co-localization of PA and fibrillarin during mitosis. If anything, the fluorescent signals generated by the mitotic cells appears to have saturated the camera.
Minor Issues
- There are numerous grammatical errors made throughout the manuscript that need to be addressed to improve the clarity of the manuscript. In addition, the authors need to shorten their numerous run-on sentences.
- The authors need to be better at providing references for the statements that they make in this manuscript.
- The Materials and Methods section seems rather disorganized. For example, why is a description of their wide-field microscopy system provided in section 2.1 “Cell lines, cell culture, and transfections”? In addition, why is information about plasmid DNA purification provided in section 2.2 “Structured illumination microscopy”?
- The authors should state which oil they used for obtaining their DeltaVision OMX images.
- What is the light source that the authors used for their wide-field microscopy?
- The authors need to identify the companies from which they purchased the chemicals and reagents used in this work.
- Which secondary antibody did the authors use for their Western blots and where did they purchase it form?
- Gene names are written in italics.
- All abbreviations need to be defined. However, there is no need to keep defining an abbreviation after it is first defined.
- What is the “BC buffer” described in section 2.3 of the Materials and Methods?
- What do the authors mean by “disks discarded” in line 191 of page 5?
- In the 1st sentence of section 2.17 of the Materials and Methods, did the authors mean to write “genomes” instead of “proteomes”?
- The authors need to do a better job of explaining what TGB1 is beyond calling it a “viral protein”. Which virus does it come from and why is it interesting to consider in relation to fibrillarin?
- The authors need to explain why they decided to test the effect of Ca2+ on the ribonuclease activity of fibrillarin in Figure 2C, 3B, and 5D?
Author Response
Major Issues
The majority of the results presented in this work are not quantified, which makes it difficult to assess their rigor and reproducibility. Moreover, there are no statistical tests performed to evaluate the significance of the authors’ results. This is simply not acceptable in this day and age.
We carried several times all the experiments. The quantified values are now indicated in the figure legends. We added the quantitative measurements into the representative gels and the corresponding data on the FRAP assays and supplementary quantitative information from the fibrillarin ring, size and statistical measurements. Supplementary figure 3. The corresponding information shown in the microscopy images are indicated and correspond to all of the cells that were observed in ten independent experiments in which all non-mitotic cells showed the same fibrillarin-PIP2 pattern. We added an additional figure in the Supplementary figure 4 showing different cells stained with different antibody, but maintain the same fibrillarin-ring structure and their measurement is not changed. Also we added Supplementary figure 6 to show colocalization of SNAP-FIB with GFP-FIB used in the frap experiments.
The Materials and Methods section is rather disorganized and does not clearly describe how many of the experiments presented in this manuscript were performed. Several examples are provided below.
To address this issue, we put a great effort to improve the structuration of the method section of the manuscript, in particular section 2.1. entitled “Cell lines, cell culture and transfection assays” and section 2.2. entitled “Microscopy”.
In section 2.8, the authors describe the use of the “in vitro labeling with SNAP-Biotin strategy” for labeling their native fibrillarin. However, the authors do not describe having ever tagged fibrillarin with the SNAP-tag. They go on to state that they used Biotin-streptavidin beads to purify fibrillarin from cell lysates without identifying which cells were used. Thus, can the authors really say that they are purifying “native fibrillarin”, as the title of section 2.8 suggests?
We apologize and it is a good point to define this section properly. We reorganized the section 2.1 and describe the generation of stable SNAP-Fibrillarin cell lines and how it was used to purify the native complex using SNAP-Fibrillarin as a tagged protein. We also changed the title of section 2.8 to: “SNAP-Fibrillarin purifications from HeLa cells”. As detailed in this section the purification was carried out by streptavidin-magnetic beads from Thermo Fisher Scientific after feeding the cells with SNAP-Biotin.
The Materials and Methods does not mention anything about the use of a fluorescent SNAP-tag-binding ligand. Thus, I am confused as to what exactly the authors did to generate the image show in Figure 2E.
As requested by Reviewer 2, novel section 2.1 was included to clarify the SNAP-Fibrillarin stable cell line generation, in particular, lines 113 to 124. Figure 2e allows the visualization of SNAP-fibrillarin and PIP2 signals, in which the red signal corresponds to PIP2 and the green signal corresponds to SNAP-fibrillarin. The stable cell line generation is detailed in section 2.1, starting as: “Stable SNAP-Fib cell line was generated as follows: transfection with PEI was performed in an 80% confluence 60mm plate of U2OS osteosarcoma cell line....”. We also compared the SNAP-fibrillarin cell line to the GFP-fibrillarin in the same cell to prove that they localize in the same region as shown in the new Supplementary figure 6.
In section 2.16, the authors do not explain how they were able to select a region of interest for performing their FRAP experiments. Since the DeltaVision OMX is not a confocal microscope, this information is necessary for the reader to be able to critically assess the FRAP experiments presented in Figures 5E-F. Also, what do the arrows and arrowheads signify in Figure 5E?
We apologize for the lack of details regarding the selection of regions of interest, which are now provided in 2.16 section lines 298-299. Also, figure 5f shows representative Nucleoli and Cajal bodies selected for photobleaching marked in red circles. Recovery times and their images were taken at 60 and 100 sec, as shown in figure 5f.
The omission of description for the arrows and arrowheads was corrected and indicate that they point to Nucleoli and Cajal bodies, respectively. We also added the p values and statistical data for these experiments.
How many cells were the 200 independent FRP measurements of WT and mutant fibrillarins performed in?
We clarified this point indicating that one independent laser photobleaching corresponds to one single cell i.e., photobleaching one Cajal body or one nucleolus. Therefore, it was 200 independent cells during different days. Since the cells could only be photobleached once to avoid unreliable data due to damage of the cells. The regions of interest were selected manually as stated in materials and methods, section 2.16. The figures now show the bleached area and its recovery over time. Also, we show the statistics from the experiments and the n values for the SD.
How were the “RNPs with or without the GAR domain sequence” assembled? While section 2.7 of the Materials and Methods describes the expression and purification of these RNPs, the information is scant at best.
To avoid confusion, we termed RNP the Fibrillarin, Nop56, Nop58, 15.5-kDa protein complex with full-length fibrillarin and RNP DGAR for the same complex but with a truncated fibrillarin from amino acids 1 to 82. The sections 2.5 and 2.6, in the present manuscript, describe how the GAR domain from fibrillarin was introduced into the plasmid constructed by [40]. While the section 2.5 describes the PCR reactions, termed as Exponential megaprimer PCR (EMP); followed step by step to amplify the GAR domain coding region and the subsequent introduction into the plasmid; the section 2.6 describes the phosphorylation, ligation of the new synthesized whole DNA plasmid, and the depletion of the bacterial methylated template plasmid by DpnI enzyme. A current overview of the strategy used to introduce long sequences into plasmids is available in reference [39]. The system that is used expresses all the proteins in the cell with one of the protein tag to pull the complex as previously published [40].
How did the authors fluorescently label PIP2 in the SIM image shown in Figure 2E? The probe does not seem to be described anywhere in the manuscript.
We apologize for this omission. We used a specific primary antibody anti-PIP2 from Echelon™ (Z-A045) and goat secondary antibody anti-mouse IgM conjugated with Alexa Flour 555 from Life Sciences for PIP2 immunofluorescence. Corrections were made in lines 121 to 124 in section 2.2.
What is a “RGG-box1” model?
Again, we apologize for the missing information pertaining this term. This sentence refers reference to the glycine-arginine repeats founded in other RNA processing proteins and term as the RGG-box. Hidden Markov Models were constructed based on these repeats, and the corrected form of the previously written concept is giving lines 593-595 as: “HMM for RGG-box 1 and RGG-box 2 repeats from GAR1 proteins were constructed and other GAR1 proteins were detected in the analyzed genomes …”
What is a “Boxshade analysis”?
Boxshade is an online visualizing tool for multiple sequences alignment used in this work to show the conserved RGG motif around the LMS14-A proteins analyzed in the bioinformatic section and details are available at: https://embnet.vital-it.ch/software/BOX_doc.html. This information has been added in lines 593-595.
Is the fibrillarin that the authors purified in this work functional? In other words, does the purified fibrillarin perform its established job as an rRNA or H2A methlytransferase?
This is a good question and as pointed out in the introduction only archaea fibrillarin has been shown biochemically to methylate rRNA. This mostly results from the difficulty to obtain recombinant fibrillarin to perform the methylation rRNA experiments. We were interested in studying the guide RNA interactions with fibrillarin when we unexpectedly discovered the ribonuclease activity of this protein with other RNAs. In our previously work [13] we were able to reproduce the methyltransferase activity of recombinant fibrillarin on the glutamine 106 in histone H2A, arguing for a good quality of the protein after purification.
The authors show a plot of the results of their FRAP experiments in Figure 5F without providing any images to show how or where the photobleaching was performed in cells expressing the indicated fibrillarin-EGFP constructs. These images are necessary for the reader to critically evaluate, not to mention believe, these experiments.
As recommended Figure 5f was modified in order to introduce images pointing the selection of Nucleoli and Cajal Bodies showing complete photobleaching and recovery of the specific section. A better description and more details were added in section 2.16.
The authors conclude from their single SIM image shown in Figure 2E that in nucleoli “the fibrillarin surrounded PIP2 in the DFC and the ring-like structure involved a change of phase which outside the DFC lacks the lipidic environment”. However, I cannot see how they are able to conclude anything about the phase transition of fibrillian from a single static image.
We apologize for not properly introducing the phase transition and dynamics of fibrillarin, which have been well study. Indeed, among other approaches, several laboratories investigated this concept by using different time frame microscopy or microfluidic testing. Here we wanted to show the ring-like structure that fibrillarin makes around PIP2, which is novel. PIP2 is a well-studied phospholipid, and much has been published about its role in lipid islets or in speckles. However, nucleoli lack any diffuse PIP2 and only show a particular set of spots. We find interesting how fibrillarin surrounds this lipid considering the dynamics for which fibrillarin is known for. Further testing should be carried out to define how this structure behaves under different situations.
I would caution the authors from making drawing too many conclusions from the images presented in Figure 3E regarding the co-localization of PA and fibrillarin during mitosis. If anything, the fluorescent signals generated by the mitotic cells appears to have saturated the camera.
We completely agree with the reviewer and a statement was added to clarify that during mitosis the level of PA increased and that nuclear membrane fragmentation leads to alteration of the majority nuclear structures. Because of the increased signal is impossible to define if there is a colocalization. However, the observation of an increased level of PA in the proximity of fibrillarin is in agreement with a scenario of an interaction of fibrillarin with PA. Previous reports have shown PA pulling down fibrillarin from extracts, together with the Fat blots provide the reason to test the effect of this lipid on this assay. The inhibitory effect on the ribonuclease activity as well as the release of guided RNA in the presence of PA was unexpected.
Minor Issues
There are numerous grammatical errors made throughout the manuscript that need to be addressed to improve the clarity of the manuscript. In addition, the authors need to shorten their numerous run-on sentences.
We apologize for this issue. To address this aspect, the manuscript was edited by a native English speaker and editor.
The authors need to be better at providing references for the statements that they make in this manuscript.
Accordingly, more concise referenced sentences were added to the manuscript in order to support our own hypothesis or ideas.
The Materials and Methods section seems rather disorganized. For example, why is a description of their wide-field microscopy system provided in section 2.1 “Cell lines, cell culture, and transfections”? In addition, why is information about plasmid DNA purification provided in section 2.2 “Structured illumination microscopy”?
In the new version of the manuscript, we restructured this section and provided the pertinent corrections as follows in the downstream points. Section 2.1 states the transient and stable transfections made in HeLa and U2OS cells, while the section 2.2 details all the microscopy tools used and strategies followed in the study.
The authors should state which oil they used for obtaining their DeltaVision OMX images.
As requested by reviewer 2 the Oil brand used is stated in lines 129 of the 2.2 section. We also indicated in the microscopy section the refractive index.
What is the light source that the authors used for their wide-field microscopy?
Light source was added to the 2.2. Microscopy section: Leica EL6000 with HXP 120W / 45C VIS Hg lamp for fluorescent lamp light source was used, lines 128-129.
The authors need to identify the companies from which they purchased the chemicals and reagents used in this work.
This has been corrected throughout the manuscript.
Which secondary antibody did the authors use for their Western blots and where did they purchase it form?
As requested by reviewer 2 we describe the secondary antibody used in lines 257, using the anti-mouse IgG secondary antibody from GE Healthcare, as described in materials and methods, section 2.9.
Gene names are written in italics.
Style per Gene names was corrected as follow i.e. FIB and hGAR1 and a reference was also indicated.
All abbreviations need to be defined. However, there is no need to keep defining an abbreviation after it is first defined.
This has been corrected throughout the manuscript.
What is the “BC buffer” described in section 2.3 of the Materials and Methods?
The commonly use BC-100 buffer is now defined in the section 2.4., with the corresponding composition and its abbreviation, lines 173.
What do the authors mean by “disks discarded” in line 191 of page 5?
We apologize the syntax error, the sentence should be: “The media from the HeLa cell cultures was discarded and the cell monolayer washed…”, line 238, section 2.8.
In the 1st sentence of section 2.17 of the Materials and Methods, did the authors mean to write “genomes” instead of “proteomes”?
Indeed “proteomes” was meant.
The authors need to do a better job of explaining what TGB1 is beyond calling it a “viral protein”. Which virus does it come from and why is it interesting to consider in relation to fibrillarin?
We address the suggestion from the Reviewer in order to make clearer the introduction and experiments performed with the viral protein TGB1. As stated in lines 603-610, some movement viral proteins possess nucleolar signals to direct their location to the nucleolus[61], these regions contain IDR that also interacts with fibrillarin[62]. Therefore, the ribonuclease activity of TGB1 was evaluated as a control and to check if their IDR could possess RNAse behavior.
The authors need to explain why they decided to test the effect of Ca2+ on the ribonuclease activity of fibrillarin in Figure 2C, 3B, and 5D?
To address this issue, we now state in lines 382-386 as: “Calcium ions have been detected in the crystal structure of human fibrillarin and other enzymes with ribonuclease activity are either dependent on or inhibited by calcium. Our previously work showed that the AtFib2 from A. thaliana is activated with a small amount of calcium in ribonuclease activity assays, but no effect was observed in the same context with AtFib1 enzyme”. Following these observations, we tested the influence of calcium on the activity of the human enzyme fibrillarin as described in the section 3.1., 3.2., and 3.3. in figures 2c, 3b and 5d.
Reviewer 3 Report
In the manuscript Fibrillarin ribonuclease activity is dependent on the GAR domain and modulated by phospholipids, the authors work to classify the novel ribonuclease activity of fibrillarin, including looking at a number of activators and inhibitors of this activity and investigating the role of the GAR domain in facilitating these activities. Unfortunately, it is the opinion of this reviewer that the paper is not yet ready for publication. The main concern lies in how unapproachable the manuscript is. So much information is missing in the introduction, figures, and figure legends. These omissions have made it difficult, and in some cases impossible, to make meaningful observations about the scientific validity of this paper.
Below I have provided specific feedback, but will once again stress the need for additional information and explanation to make this manuscript fit for publication. In its current form it feels like the bones of a paper, one that would benefit greatly from being fully fleshed out.
Point-by-point concerns follow:
Major concern
The introduction is very bare and does not adequately introduce the concepts necessary for approaching this paper. Key components of pathways are brought up without proper introduction or background. Similarly, the discussion seems as though it could be expanded to better comment on the relevant results of these experiments.
Figure 1f, g, h: These figures are not well explained, either in text or in the figure legend. First, in figure f, what is the ADN band? It’s that meant to be DNA? If so, why is there a DNA band? Your explanation mentions only RNA. What is the 28S and 18S band: assumedly RNA, why not state this directly? In the text you mention 5S RNA, where is this? There isn’t any on the gel. To my eye, the 28S band looks like it is already undergoing degradation in the control lane: this is problematic for a control.
What is the relevance of Ca2+? Please explain your reasoning behind selecting this for addition into your experiment.
g&h: Any reason you didn’t include a + Ca, + Ptd, + Fibrillarin lane?
Figure 2: a) What is Nop58, and Nop56? This is not explained. Why is 15.5K labeled? What is the relevance?
- b) No concerns
- c) There seems to be loading issues because in the Ca+ lanes, the bands are lighter, making it hard to analyze whether or not there was, in fact, more degradation.
- d) No concerns
- e) Neither figure is exactly an inset. I’m assuming that the right image is a zoomed in version of the left image?
Moderate concern
Figure 1b. You need to do a better job explaining why there are two bands here. You address it in the text but only in a roundabout manner. State what the smaller band is, why you think that, and the relevance behind there being two bands.
Figure 2: You describe RNP as removing the first 81 nucleotides, and then adding them back in. Am I missing something, or is this just full length RNP? Why describe it as a deletion and insertion?
Figure 3A: what is the moderate change that PtdIns(4,5)P2 caused? They appear the same.
3B: No concerns
3C: This figure is not self-explanatory with the legend alone. What do S, W, and A mean?
3D: No concerns
3E: Need to state what your colors represent in the figure legend.
Figure 4: This figure is better, but you begin talking about BCO without any introduction as to what it is or why it’s relevant.
Figure 5: You mention that R45A degrades 28S rRNA better, but never fully discuss why this might be in the Discussion. Why is it more active than wildtype? Why might it be more active than the R34A mutant?
Figure 6: Similarly, you should expand on what Figure 6d is indicating. What is the relevance of the different activity of the different GAR domains? Is there a relevant pattern?
Line 45, you mention H2A without any context for what it is or does.
PtdIns(4,5)P2, PIP2 not introduced properly
Line 322: The abbreviation PA appears. Introduce your abbreviations in text, not just figure legends.
Author Response
Major concern
The introduction is very bare and does not adequately introduce the concepts necessary for approaching this paper. Key components of pathways are brought up without proper introduction or background. Similarly, the discussion seems as though it could be expanded to better comment on the relevant results of these experiments.
Following these suggestions, we incorporated in lines 39 to 56 the ideas and concepts to adjust properly the manuscript to the journal scope, and also to make clearer the relevance of the different components used to present each result. For instance, we added in the introduction the relevance of the nucleolus and the plethora of internal nuclear regulations in order to produce functional ribosomes. We also introduced the concept of liquid-liquid phase separation. In the results section we defined the ribonucleoparticle concept, the use of calcium ion, RGG repeats and their relevance, just to cite a few examples. In the discussion section, we introduced some context about ribosomal biogenesis, phase separations, other molecules binding fibrillarin and the complexity to address these concepts in a whole context, such as for biogenesis of rRNA and their processing steps. We also explained in a greater manner the importance of other molecules that regulate fibrillarin function, localization and behavior during some of these related processes. Finally, we also discussed the influence of PTM in the GAR domain section of fibrillarin, and their relevance to regulate the whole methylation activity of fibrillarin during the cell cycle.
Figure 1f, g, h: These figures are not well explained, either in text or in the figure legend. First, in figure f, what is the ADN band? It’s that meant to be DNA? If so, why is there a DNA band? Your explanation mentions only RNA. What is the 28S and 18S band: assumedly RNA, why not state this directly? In the text you mention 5S RNA, where is this? There isn’t any on the gel. To my eye, the 28S band looks like it is already undergoing degradation in the control lane: this is problematic for a control.
We addressed the inquiries about the proper description on Figure 1. The DNA band described in Figure 1f corresponds to the DNA co-purified using the TriZol protocol, as DNA co-purification under similar conditions was previously reported in the literature. We considered that DNA should not alter the relevance of the activity of fibrillarin in our results and therefore performed experiments. Corrections were made in order to make clearer these points. Also, we described and pointed properly that the bands that appear in the lane 2 from Figure 1f are the major populations of 28/18S rRNA, which is now described in figure legend and results. The mention “The 28S rRNA was mainly affected but not 5S rRNA” reports observations appreciated only on early stages on the agarose electrophoresis, suggesting a preference of fibrillarin to degraded some particular rRNA population and “Data not show” was added to the text. High-quality RNA purification was used in the Figures 1f, g, and h, and even if some degradation can be seen it in these figures (control lines), the mayor degradations correspond to the addition of fibrillarin to the in vitro assay. Note that the controls were submitted to the same time and temperatures used for the incubations and no degradations were observed.
What is the relevance of Ca2+? Please explain your reasoning behind selecting this for addition into your experiment.
Also referred by Reviewer 2, we addressed the use of calcium because calcium ions have been described as cofactors for some ribonucleases as a general rule of activity, as long as some ribonucleases are activated or inhibited in presence of calcium. We decided to address the influence of calcium in regard to our previously published work on AtFib2 showing an in vitro effect [50], wondering of the occurrence of a general influence of calcium on the activity of human fibrillarin.
g&h: Any reason you didn’t include a + Ca, + Ptd, + Fibrillarin lane?
According to the suggestion we properly labeled figure lines with +Ca, +Ptd, +Fib in the new Figures 1g and 1h.
Figure 2: a) What is Nop58, and Nop56? This is not explained. Why is 15.5K labeled? What is the relevance?
The context regarding RNP was not properly introduced leading to some misunderstanding. We introduced the reason to use this complex and its relevance in the methylation conducted by fibrillarin during rRNA processing. The expression vectors used in this study are described in Material and methods, sections 2.5., 2.6., and 2.7. The purification under native conditions assumes the copurification of the four proteins that compose the complex taking in mind that are self-interacting in vivo, also tested in [40] with results similar to ours. This is the reason to label the other three components in the SDS-PAGE gel.
- b) No concerns
- c) There seems to be loading issues because in the Ca+ lanes, the bands are lighter, making it hard to analyze whether or not there was, in fact, more degradation.
We agree with the minor reduction seen in both conditions with recombinant protein or in the full complex, does not alter the pattern in which RNA is cut, and if there is an effect it is minor. Indeed, analysis of 10 repeats showed in all cases a small reduction in the lane but no alteration in the pattern in which the RNA is cut. Therefore, the role of calcium in this activity is minor compared to the plant fibrillarin 2 that was previously reported to be activated by calcium[50].
- d) No concerns
- e) Neither figure is exactly an inset. I’m assuming that the right image is a zoomed in version of the left image?
This is correct and we have marked the section that was zoomed in order to show the particular ring structures that we commonly observed with fibrillarin. We also added quantitative measurements in the Supplementary figures 3 and 4.
Moderate concern
Figure 1b. You need to do a better job explaining why there are two bands here. You address it in the text but only in a roundabout manner. State what the smaller band is, why you think that, and the relevance behind there being two bands.
Recombinant proteins, in particular fibrillarin, are well known to be prone to degradation, even in the constant presence of protease inhibitor cocktail. So, we assume that the lower bands detected in the Figures 1a (line 2), 1b, 1c (lane 2) are products of protein degradation with some ribonuclease activity as depicted in the Figure 1c. This was also observed in the experimental strategy used in [62] to demonstrate the interaction between the viral protein ORF3 and fibrillarin.
Figure 2: You describe RNP as removing the first 81 nucleotides, and then adding them back in. Am I missing something, or is this just full length RNP? Why describe it as a deletion and insertion?
Associating the results from the figure 1b and 1c, we speculated that GAR domains or at least a segment in the N terminal domain could be important for the ribonuclease activity observed in fibrillarin alone. We decided to test if the activity is present in the context of an RNP complex, as stated in, section 3.2. We used a previously constructed plasmid for recombinant expression of the four proteins as reported by [40]. The authors used the construct to produce functional complexes to test the methyltransferase activity, and as noted in [10] the methyltransferase domain and the S-adenosylmethionine binding site are in the C-terminal region of the protein. The coding sequence of fibrillarin in the construct from [40] did not contain amino acids 1-82, corresponding to the GAR domain, and termed this construct as RNP DGAR construct. In the methodology sections 2.5. and 2.6. we introduced the GAR domain coding sequence, terming this construct as RNP. Long DNA sequence introduction into plasmids was followed as described in [39].
Figure 3A: what is the moderate change that PtdIns(4,5)P2 caused? They appear the same.
The change in the mobility of the band is a small shift and may imply a conformational change. We added a specific figure in the supplementary figure 5D section that describes this observation with more details in a different experiment.
3B: No concerns
3C: This figure is not self-explanatory with the legend alone. What do S, W, and A mean?
Correction on the paragraph was made describing S for silver staining, W for western blot and A for activity assay, also described in the figure legend.
3D: No concerns
3E: Need to state what your colors represent in the figure legend.
We now provide a description of the colors used in the micrograph, yellow for phosphatidic acid and red for fibrillarin, the merge is presented in the right panel of figure 3c.
Figure 4: This figure is better, but you begin talking about BCO without any introduction as to what it is or why it’s relevant.
Certainly, we introduce the BCO concept without any previous context, as Reviewer noted. In order to correct this we added the relevance of BCO sequence, detailed in lanes 476-477 and reference [10].
Figure 5: You mention that R45A degrades 28S rRNA better, but never fully discuss why this might be in the Discussion. Why is it more active than wildtype? Why might it be more active than the R34A mutant?
As the Reviewer kindly suggested, we included a possible explanation for the increased activity of fibrillarin mutants. As far as we know GAR domain regions are the principal mediator of lipid-binding and mutations into two conserved sites could lead to a conformational change that may modify lipid affinity. It is interesting to note that arginine residues are commonly found key residues in PA interaction motives.
Figure 6: Similarly, you should expand on what Figure 6d is indicating. What is the relevance of the different activity of the different GAR domains? Is there a relevant pattern?
As suggested expanded information was added to explain why we have tested the ribonuclease activity of other GAR domains found in other RNA-binding proteins. We speculate that some degree of degradation could be observed, and the figure 6d describes that, in part, our hypothesis is correct. Alternatively, for TGB1, a protein that possesses another type of IDR shows no observed effect in the in vitro activity assay. This negative control shows that no all the disordered regions possess ribonuclease activity and that the same buffers for purification have no activity in them.
Line 45, you mention H2A without any context for what it is or does.
We described in lines 59-61 of the new version, the relevance of methylation H2A as an epigenetic mark present in human and yeast organisms.
PtdIns(4,5)P2, PIP2 not introduced properly
This is done line 75-76.
Line 322: The abbreviation PA appears. Introduce your abbreviations in text, not just figure legends.
PA abbreviation is first mentioned on line 98.
Round 2
Reviewer 2 Report
My responses are shown in blue text. I also suggest that the authors modify Figure 1D by labeling specific amino acids in the binding sites indicated by the arrows in this figure.

Author Response
Reviewer 2
We are very grateful for this reviewer as it help us improve the manuscript. Again in Bold are the Reviewer comments followed by the way it was solved
The “computational inference of possible binding sites for phospholipids” shown in Figure 1D really needs more information. Specifically, the authors need to label specific amino acids in the binding sites indicated by the arrows in this figure.
We added in figure 1D the amino acid sequence that indicates the phospholipid binding sites as requested.
Major Issues
The authors should really quantify the results of the fat blot assays shown in Figures 1E and 5C. How many times were these assays repeated? Can statistical tests be performed to determine the significance of these results?
I appreciate the fact that the authors now show the average (I assume that it is the average, as it is not explicitly stated in the Figure legend) levels of 28S and 18S below the plots shown in Figures 1E, 1F, 1G, 2B, 2C, 2D, 3F, 4D, 4F, and 5D. However, it would really be better if the authors could display these results in plots. The authors should also perform l tests to determine if the differences observed between the experimental conditions tested in each experiment are significant or not.
The Fat blot assays from 1E and 5C were duplicates preventing us to perform a proper statistical analysis. Nevertheless, the average values and standard deviation are now indicated in the figures. Supplementary figure 1 details the quantification results from figures 1E and 5C.
We replaced the lower tables from all the experiments and added plots as requested, and significance was assessed using statistical t test, as described in figures 1E, 1F, 1G, 2B, 2C, 2D, 3F, 4D, 4F, and 5D.
Unless the authors generated a HeLa cell line in which they used CRISPR/Cas9 to fuse the cDNA encoding the SNAP tag with the endogenous FBL gene such that said cell line expresses SNAP-tag-fibrillarin, they cannot say that they purified native fibrillarin.
We agree and changed the terminology to the correct term as HeLa expressed and purified SNAP-fibrillarin in line 239
How do the authors know that the fibrillarin-GFP structures observed in the images of nuclei shown in Figure 5E are Cajal bodies or nucleoli based only on GFP fluorescence?
A bibliographical reference supporting the claim that the structures seen in the transient expression of GFP tagged Fibrillarin in figure 5 are nucleoli and Cajal bodies. Considering that cellular Fibrillarin localization has been well establish in the field and that we previously determined that native fibrillarin colocalized with UBF and RNA pol I (25,26), we are confident that GFP tagged fibrillarin is associated with nucleoli and Cajal bodies. To further support this assertion, we added Supplementary figure 6, where we investigated colocalization of anti-fibrillarin and SNAP-fibrillarin and the GFP tag fibrillarin together with SNAP-fibrillarin. In all cases the main signal colocalized in the same locations. Of note, the localization of fibrillarin mutants used in the FRAP study was not affected but their dynamics clearly was modified.
Also, what do the error bars shown in the plots in Figure 5E represent? Standard error? Standard error of the mean? Furthermore, it would be good if the authors included in the Figure 5 legend a description of the p values represented by the “*”’s in the plots shown in Figure 5E.
We apologize for this mistake. Error bars represent the standard error of the mean (SEM). This information is now included in the figure 5E.
Moreover, I am confused by the plot of “FRAP dynamics normalized from 200 independent photobleaching events for WT and mutant fibrillarins in CBs and Ncos” shown in Figure 5F. Do each of the curves shown in this plot represent the average fluorescence recovery for a given construct over time? If so, the authors should include error bars for each data point shown in the plot.
The curves in figure 5f represent the normalized values of the mean of the 200 repetitions for each condition and for each time point. In fact, means values where normalized to 1 in order to obtain a simple, smooth comparable curve that would allow the reader to make a qualitative assessment of the dynamics changes between each mutant in Nucleoli and Cajal bodies, compared to WT fibrillin. This is more clearly stated in the figure legend. Because of the normalization of the values, adding the SD bars would over saturate the figure and make proper reading impossible due to an excessive number of lines.
Why does the Diffusion Coefficient plot shown in Figure 5F lack error bars and statistical tests?
The diffusion coefficient is a single value that is obtain through the next formula:
Take note the formula appears in the PDF file added
Where r2n corresponds to the ROI radius and τ1/2 is the mean obtained from the 200 FRAPs repetitions. This gives us one single representative value of the diffusion speed for fibrillarin and its mutants in individual structures (CB and Nco) and allows us to compare the effect of the point mutations over Fibrillarin dynamics in a quick and practical way. Obviously, this prevents to add error bars or statistical tests, but the Half-life data were also included to support the analysis.
4) The authors show a plot of the results of their FRAP experiments in Figure 5F without providing any images to show how or where the photobleaching was performed in cells expressing the indicated fibrillarin-EGFP constructs. These images are necessary for the reader to critically evaluate, not to mention believe, these experiments.
- Thank you for providing this clarifying information.
- However, it might make more sense to show the representative images of photobleaching over time before showing the plots that quantify the results of these experiments. This suggestion would be applicable to both Figures 5E and 5F.
The figure 5 was reorganized following the reviewer’s suggestions.
Thank you for the clarification. However, does a simple cellular assay for fibrillarin that they authors can use to demonstrate that their fibrillarin constructs are functional? In other words, can the authors rescue the effect of depleting fibrillarin from HeLa cells with the fibrillarin constructs used in this work? For example, the authors use a GFP-tagged fibrillarin construct that is referred to as “fibrillarin-GFP” indicating that GFP is fused to the C-terminus of fibrillarin.
They also use a SNAP-tag-tagged fibrillarin construct that they refer to as “SNAP-fibrillarin” indicating that the SNAP-tag is fused to the N-terminus of fibrillarin. A SNAP-tag is 19.4 kDa and GFP is 27 kDa. Since the fusion of labels of the size of SNAP-tag or GFP to the N- or C terminus of protein can have deleterious effects on the function of said protein, it is possible that the SNAP-fibrillarin and fibrillarin-GFP constructs used in this work are not functionally equivalent, despite the fact that they show a similar localization pattern in cells.
This is a very good comment and by no means a trivial issue to solve. However, to support the notion that both tagged fibrillarins are functional, we added two new supplementary figures parts. One with cells that expressed the GFP-fibrillarin mutant K131E,A250P, as Tollervey and colleagues previously published that such amino acid substitution affected the methylation function of fibrillarin resulting in cell death (67). The GFP fibrillarin mutant K131E,A250P, expressed in the cell line SNAP-fibrillarin shows an alteration in cellular localization of the mutant fibrillarin but not of the wild type SNAP-Fibrillarin as seen in the novel Supplementary figure 6. Therefore, this nonfunctional fibrillarin shows a different pattern in the cell. Alternatively, the distribution of stably expressed SNAP-fibrillarin is the same as the native fibrillarin as anti-fibrillarin antibody staining perfectly overlaps with that of SNAP-fibrillarin (Supplementary figure 7). Furthermore, unlike some fibrillarin mutants that result in cell death, the expression of SNAP-fibrillarin is not affecting cell viability, supporting the notion that SNAP-fibrillarin is functional. In conclusion several mutations affecting the function fibrillarin result in an alteration of their localization, whereas SNAP-fibrillarin and fibrillarin-GFP colocalized with native fibrillarin, in line with the notion that these fusion proteins are functional.
Thank you for the clarification. I would suggest that the authors show single channel images of the PIP2 and fibrillarin fluorescence in addition to the merged image. In addition, the authors might consider using a pair of colors for their merge image that are friendlier to color-blind individuals. That being said I would strongly caution the authors against drawing too many conclusions from this OMX image, as fine hexagonal repeating pseudo-structures referred to as “honeycombs” in areas of increased out-of-focus signal are quite commonly found artifacts in 2D-SIM images. To address this possibility, the authors should perform 2D-SIM on cells stained for other nucleolar proteins to see if they also exhibit this localization.
We agree and improved the image to show each channel in a friendlier and easy for color-blind individuals. Thanks to this improvement we also addressed the comment for hexagonal repeating or “honeycombs” which are smaller in size compared to the fibrillarin rings and can be seen in the background image 2e, as well as in the background of Supplementary figure 7. Considering that the fibrillarin rings have a diameter of over 400 nm we can see them even in confocal microscopy, although SIM definitely improves the image.
Thanks to the friendlier color and channel separation, the PIP2 channel can now be observed showing a NON “honeycomb” pattern in the location of the center of the rings. This shows a structure similar to that seen when staining SLERT (71) and is therefore unlikely to represent an artifact.
We also added a series of Z stacks of 0.125µM in Supplementary figure 5
In addition, the authors should perform 2D-SIM in cells stained for fibrillarin and other lipids to show that they do not accumulate within the center of the observed fibrillarin ring.
We added novel data showing PA distribution in 2D-SIM together with FIB Supplementary figure 8A.
We also added an anti-PI4P staining, which is much weaker to that of PIP2 antibody. Supplementary figure 8B shows an absence PI4P signal in the center of the fibrillarin rings. However, this antibody is gives very week staining in our hands and there is very little information compared to PIP2 antibody staining in cells.
Thank you for the clarification. However, I would still discourage the authors from concluding that “the observation of an increased level of PA in proximity of fibrillarin is in agreement with a scenario of an interaction of fibrillarin with PA”. If the authors were to stain the same cells shown in Figure 3E for soluble tubulin instead of PA, a similar “proximity” to fibrillarin argument could be made. Therefore, I do not see the benefit of including this data in the manuscript.
It is indeed difficult to test this hypothesis, as during metaphase several proteins show this pattern and the conclusion is not supported by data at this time. This is clearly an issue that we want to further investigate as it may hold clues with regard to several proteins that bind phosphoinositides in the nucleolus can also bind PA. We agree to this point and such a conclusion is no longer found in the new version of the manuscript.

Reviewer 3 Report
The authors have put a commendable amount of effort into editing the manuscript to address the concerns raised by the reviewers; each of my previous points were addressed. My main issues were in approachability of the paper, which has been significantly raised by putting in a more thorough introduction, conclusion, and explanation of figures in the appropriate locations.
As a result of these additions, there are now some areas where grammatical editing would help to improve the paper. However, these issues are minor and can/should be corrected with some minor editing.
The only other issue I see worth pointing out is that in Figure 1f, your DNA band is still labeled as "ADN."
Author Response
The only other issue I see worth pointing out is that in Figure 1f, your DNA band is still labeled as "ADN."
We have revised the manuscript again for typos and errors like this. Figure 1f the DNA labeled was corrected.
Again, we thank the reviewers for all the comments made through this process.
Round 3
Reviewer 2 Report
Overall, the authors have addressed the majority of my concerns. Consequently, I feel that their manuscript is now acceptable for publication at Cells. However, I would like to briefly comment on the authors' response to my inquiry regarding the functionality of their EGFP- or SNAP-tag-tagged fibrillarin constructs. The two new supplementary figures provided by the authors in response to my inquiry do not really address if their tagged fibrillarin constructs are truly functional. The mislocalization of the K131E,A250P relative to wild type fibrillarin only shows that the presence of these mutations is sufficient to disrupt the localization of their EGFP-tagged construct. In addition, the colocalization of immunostained fibrillarin with their SNAP-tag-tagged fibrillian construct really only shows that the anti-fibrillarin antibody used in this experiment recognizes the fibrillarin protein expressed by the cDNA encoding the SNAP-tag-tagged fibrillian construct. This is because the subcellular localization of protein does not always correlate with the function of said protein. Therefore, perhaps the authors could find a citation for a paper that shows that the N-terminal tagging of fibrillan does not significantly alter its cellular function.